# Cohesin couples transcriptional bursting probabilities of inducible enhancers and promoters

Irene Robles-Rebollo [1,2,3], Sergi Cuartero [1,2,7], Adria Canellas-Socias[1,2,8], Sarah Wells[1,2], Mohammad M. Karimi [1,2,9], Elisabetta Mereu[4], Alexandra G. Chivu [1,2,10], Holger Heyn [4], Chad Whilding[1,2], Dirk Dormann[1,2], Samuel Marguerat [1,2], Inmaculada Rioja[5], Rab K. Prinjha[5], Michael P. H. Stumpf [3,6], Amanda G. Fisher [1,2] & Matthias Merkenschlager [1,2✉]

Innate immune responses rely on inducible gene expression programmes which, in contrast to steady-state transcription, are highly dependent on cohesin. Here we address transcriptional parameters underlying this cohesin-dependence by single-molecule RNA-FISH and single-cell RNA-sequencing. We show that inducible innate immune genes are regulated predominantly by an increase in the probability of active transcription, and that probabilities of enhancer and promoter transcription are coordinated. Cohesin has no major impact on the fraction of transcribed inducible enhancers, or the number of mature mRNAs produced per transcribing cell. Cohesin is, however, required for coupling the probabilities of enhancer and promoter transcription. Enhancer-promoter coupling may not be explained by spatial proximity alone, and at the model locus *Il12b* can be disrupted by selective inhibition of the cohesinopathy-associated BET bromodomain BD2. Our data identify discrete steps in enhancer-mediated inducible gene expression that differ in cohesin-dependence, and suggest that cohesin and BD2 may act on shared pathways.

[1] MRC London Institute of Medical Sciences, Institute of Clinical Sciences, Faculty of Medicine, Imperial College London, Du Cane Road, London W12 0NN, UK. [2] Institute of Clinical Sciences, Faculty of Medicine, Imperial College London, Du Cane Road, London W12 0NN, UK. [3] Theoretical Systems Biology Group, Imperial College London, London, UK. [4] CNAG-CRG, Centre for Genomic Regulation (CRG), The Barcelona Institute of Science and Technology (BIST), Barcelona, Spain. [5] Epigenetics RU, GlaxoSmithKline Medicines Research Centre, Stevenage, UK. [6] School of BioSciences and School of Mathematics and Statistics, University of Melbourne, Parkville, VIC 3010, Australia. [7] Present address: Josep Carreras Leukaemia Research Institute, Badalona, Spain. [8] Present address: IRB, Institute for Research in Biomedicine, Barcelona, Spain. [9] Present address: Comprehensive Cancer Centre, School of Cancer & Pharmaceutical Sciences, Faculty of Life Sciences & Medicine, King's College London, London, UK. [10] Present address: Department of Molecular Biology and Genetics, Cornell University, New York, USA. ✉email: matthias.merkenschlager@lms.mrc.ac.uk

C ohesin cooperates with CTCF in organising the genome into self-interacting domains that each contain a small number of genes and enhancers, thus potentially facilitating enhancer–promoter cooperation[1–6]. However, the acute depletion of CTCF or cohesin from mouse or human cell lines deregulates only a limited number of genes at steady-state, notably direct targets of CTCF promoter binding[7], and boundary-proximal genes that show altered transcriptional bursting[8], respectively. By contrast to steady-state gene expression, loss of cohesin severely disrupts inducible transcriptional programmes in macrophages[9] and neurons[10], as well as oestrogen-responsive genes[11].

Inducible genes in macrophages are paradigmatic for the core transcriptional response to microbial signals[12]. Mechanistically, the engagement of anti-microbial pattern recognition receptors drives the recruitment of transcription factors, chromatin remodellers, and cofactors, including histone acetyltransferases. BET (Bromo- and Extra-Terminal domain) epigenetic reader proteins interact with acetylated lysine residues via N-terminal tandem bromodomains termed BD1 and BD2 (ref. [13]) and recruit Mediator and RNA polymerase to initiate eRNA transcription[12,14]. Enhancer activation culminates in target gene transcription[12] by facilitating recruitment of the transcriptional machinery to promoters[14] and by increasing the frequency of promoter bursting[15]. Transcription is typically discontinuous and can be described in terms of the frequency and the size of transcriptional bursts[16–19]. Both the size[20–24] and the frequency[22,23,25–28] of transcriptional bursts have been linked to inducible gene expression in different experimental systems. How cohesin contributes to the regulation of transcriptional bursting during immune gene activation remains unknown.

Here we address the transcriptional parameters that underlie the activation of inducible genes and enhancers in macrophages and determine steps within the process that are cohesin-dependent or -independent. We use single-molecule RNA fluorescence in situ hybridisation (smRNA-FISH) to quantitatively assess the fraction and intensity of enhancer and target gene transcription as well as the coordination between the transcriptional activity of enhancers and promoters at the single-molecule level. smRNA-FISH indicates that inducible immune genes and the activity of associated enhancers are regulated primarily by an increase in the fraction of actively transcribing alleles, and that inducible enhancer activation is tightly coupled to an increased probability of target gene transcription in wild-type cells. We find that cohesin is dispensable for activation-induced increases in the probability of enhancer bursting, and that the number of mature mRNAs produced per transcribed target gene allele is largely independent of cohesin. However, cohesin is critical for coupling the probabilities of enhancer and target gene bursting. Finally, we present evidence that the role of cohesin in enhancer–promoter coupling may not be explained by spatial proximity alone and demonstrate potential convergence of cohesin and BD2, the second BET protein bromodomain, in the activation of inducible genes.

## Results

To validate the role of transcription in LPS-inducible macrophage gene expression we compared GRO-seq (as a measure of transcription) with RNA-seq (as a measure of transcript levels). RNA-seq and GRO-seq were highly correlated ($r = 0.83$, $P < 10e-16$; Supplementary Fig. 1a). This indicates that LPS-induced macrophage genes are regulated at the level of transcription. To characterise the transcription of inducible genes such as *Il12b* we adopted a high throughput smRNA-FISH approach that allows the quantification of transcript copy numbers and transcriptional bursts in thousands of individual macrophages during the first 2 h

of the response to LPS (Fig. 1a and Supplementary Fig. 1b). smRNA-FISH counts were highly consistent between replicates (Supplementary Fig. 1c). The copy numbers of *Il12b* transcripts increased with time after LPS activation (Supplementary Fig. 1d). Transcript copy numbers of *Il12b* and other inducible genes were correlated with transcript abundance as determined by RNA-seq (Fig. 1b).

**Regulation of inducible immune gene transcription**. To identify the transcriptional parameters that underlie the transcriptional induction of immune genes in response to LPS we used nascent smRNA-FISH with probes complementary to transcribed introns. Nascent smRNA-FISH captures the fraction of actively transcribing alleles and the intensity of probe signal at transcription start sites (Burst intensity, Fig. 1c). We found that the inducible expression of *Il12b* (Fig. 1d) and other inducible genes tested by smRNA-FISH (*Egr2, Prdm1, Ifnb1, Peli1* and *Sertad2*, Supplementary Fig. 2) was associated primarily with an increase in the fraction of actively transcribing alleles. We compared the fraction of actively transcribing alleles, as directly quantified by smRNA-FISH, with the burst frequency as inferred mathematically from the moments (mean and variance) of mature mRNA distribution[18,23,24,29] (Fig. 1e, left). We found good agreement, which indicates that burst fraction and burst frequency are correlated in LPS-activated macrophages. In contrast, burst intensity as quantified by smRNA-FISH poorly correlated with burst size as inferred by the moment of mature mRNA copy number distribution (Fig. 1e, right).

To ask whether the expression of inducible genes was regulated by the frequency of transcript-expressing cells and/or the level of transcripts per cell, we performed scRNA-seq at 0, 2 and 8 h of LPS stimulation (Fig. 2). LPS increased the frequency of cells that expressed inducible transcripts for each of 6 previously identified classes of inducible primary[30] (Fig. 2, Bhatt classes A1/2, B) as well as secondary response genes[30] (Fig. 2, Bhatt classes C, D, E, F). In contrast to the frequency of transcript-expressing cells, the number of transcripts per cell increased only for class C inducible genes at the early time point. Class C to F transcript numbers accumulated at the late time point (Fig. 2, right). Taken together, these data indicate that the regulation of inducible immune genes is mediated primarily by increasing the fraction of actively transcribing alleles and the frequency of cells that express inducible transcripts.

**Regulation of inducible immune gene-enhancer eRNA transcription**. Inducible immune gene expression is regulated by inducible enhancers[12,31,32]. To explore the cis-regulatory control of inducible immune genes we designed smRNA-FISH probes for inducible immune gene enhancers that were positioned between 10 and 200 kb from their target genes within the same topologically associating domains and in the same—or immediately adjacent—contact domains in macrophage Hi-C maps: the functionally validated *Il12b* HSS1 enhancer[33] and *Ifnb1* L2 enhancer[34], as well as enhancers associated with *Egr2, Peli1*, and *Prdm1* (ref. [31]). These enhancers were actively transcribed in LPS-activated macrophages (Fig. 3a and Supplementary Fig. 3, GRO-seq), showed binding of constitutive (PU.1) as well as signal-activated transcription factors (STAT2 and IRF3), and inducible acetylation of histone H3 lysine 27 (Fig. 3a and Supplementary Fig. 3, H3K27ac, STAT2, IRF3). We used smRNA-FISH to visualise transcriptional eRNA bursting of inducible macrophage enhancers (Fig. 3b). Quantification of eRNA burst parameters indicated that activation-induced transcription of inducible immune gene enhancers occurred primarily through modulation of burst fraction (Fig. 3c), not burst intensity (Fig. 3d).

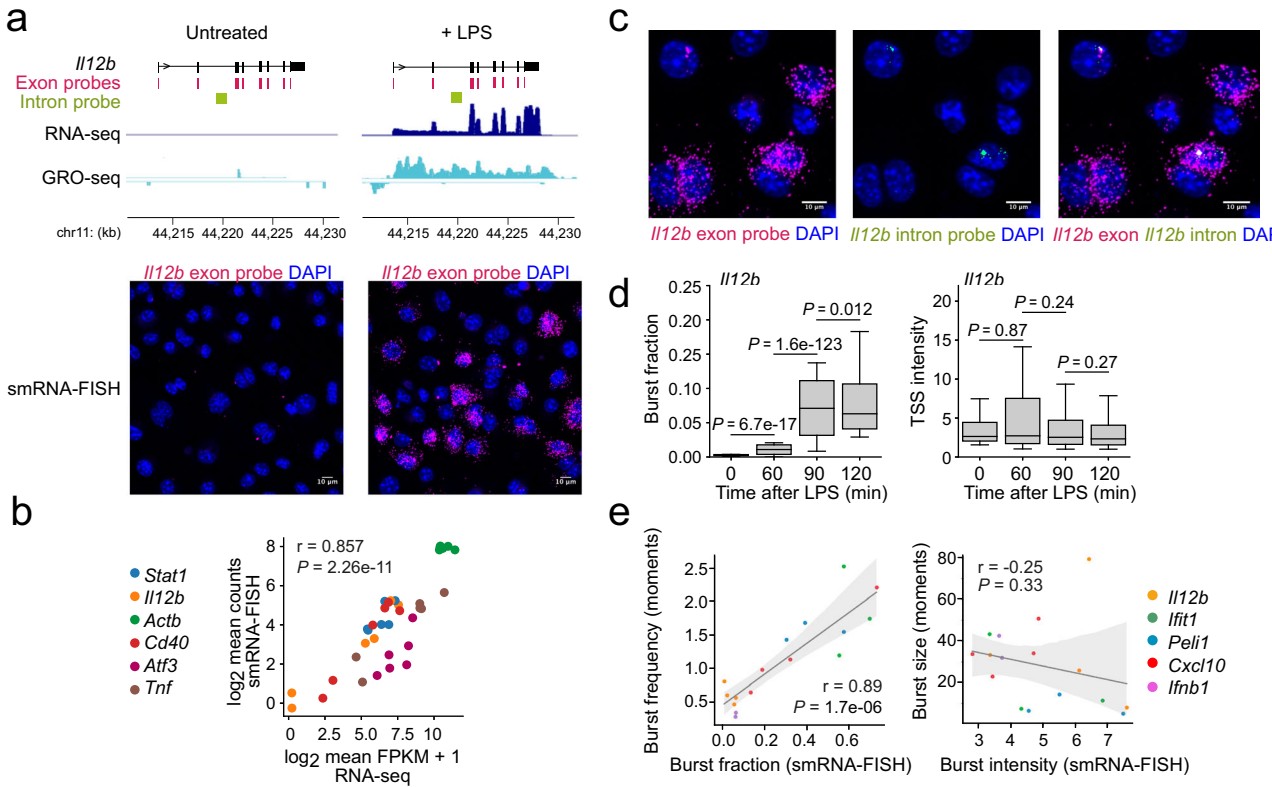

**Fig. 1 Inducible gene transcription is regulated primarily by the fraction of actively transcribing alleles. a** Induction of *Il12b* transcripts by macrophage activation as assessed by RNA-seq (top), GRO-seq (middle) and smRNA-FISH (bottom). LPS treatment times: 1 h for GRO-seq and 2 h for RNA-seq and smRNA-FISH. Representative of three independent biological replicates for RNA-seq, two independent biological replicates for GRO-seq and >3 independent biological replicates for smRNA-FISH. **b** Correlation between transcript abundance by RNA-seq (log$_2$ FPKM) and log$_2$ mean absolute transcript copy numbers by smRNA-FISH. r: Pearson correlation; P: *p* value in this and all subsequent figures. Statistical test: two-sided Pearson's product-moment correlation (Null hypothesis: correlation = 0). *Il12b, Stat1* and *Actb*: three independent biological replicates, n = 8899 cells. *Cd40, Atf3* and *Tnf*: 3 independent biological replicates, n = 12,372 cells. **c** Representative images illustrating the use of intron probes to localise transcriptional bursts. Greater than three independent biological replicates. **d** Burst fraction and burst intensity in resting and activated wild-type macrophages. In this and all subsequent box plot figures, boxes show the median and the upper and lower quartiles, whiskers show 1.5 of the interquartile range. P: *p* value. In this and all subsequent figures, we used a two-sided Tukey HSD test (ANOVA design formula ~ Sample + Replicate), adjusted for multiple testing for burst intensities and a two-sided Cochran–Mantel–Haenszel test (replicate as strata) with Bonferroni correction for multiple testing for burst fractions. N = 24,160 cells, four independent biological replicates per transcript. **e** Left: correlation between the fraction of actively transcribing alleles measured by smRNA-FISH and transcriptional burst frequencies inferred from the distribution of mature transcripts (see methods) in macrophages stimulated with LPS for 90 min (*Il12b, Ifnb1, Peli1*), 120 min (*Cxcl10*) or 180 min (*Ifit1*). Right: correlation between burst intensity measured by smRNA-FISH and burst size inferred from the distribution of mature transcripts (see methods). Intron probes were used for *Il12b and Peli1*, exon probes were used for *Ifit1, Cxcl10*, and for the intron-less *Ifnb1* gene. r: Pearson correlation. *P* value derived from two-sided Pearson's product-moment correlation test as implemented in the cor.test function in R against the null hypothesis correlation is equal to 0. N = 18,494 cells, three to four independent biological replicates per transcript. Error bands: 95% confidence interval.

**Functional coupling of enhancer and promoter bursting.** We used smRNA-FISH to visualise transcriptional bursting of both enhancers and target genes in response to LPS. Consistent with previous reports that enhancer activation can precede the transcription of target genes in macrophages[32,35] and other cell types[11,36,37], the LPS-induced transcriptional burst fraction at the *Il12b* HSS1 enhancer peaked earlier than *Il12b* mRNA (Fig. 4a, *P* = 0.0036, two-way ANOVA). We recorded eRNA and mRNA transcriptional bursts for five inducible enhancers and associated genes in a total of 38,093 cells (Supplementary Fig. 4). Burst fractions of inducible enhancers and promoters were strongly correlated (Fig. 4b). The number of cells that contained two enhancer or two promoter bursts was greater than expected based on random co-occurrence[38] (Fig. 4c), suggesting that certain cells were more likely than others to show enhancer or gene activity. smRNA-FISH probes for enhancers and their targets allowed exploration of the relationship between enhancer and promoter

bursting in individual cells, as illustrated for *Il12b* in Fig. 4d. To ask whether eRNA and mRNA bursting is coordinated, we determined the frequencies of simultaneous eRNA and mRNA transcription. Simultaneous eRNA and mRNA bursts in response to LPS activation were significantly more frequent than expected if enhancers and promoters were to burst independently (Fig. 4e). Since mature macrophages are quiescent diploid cells, the occurrence of two enhancers or two promoter bursts in a single cell indicates that bursts occur on different chromosomes in trans. The great majority (>95%) of such bi-allelic bursts were >1 μm apart (Fig. 4f). We, therefore, used 1 μm as a threshold below which we could assign enhancer and promoter bursts to the same allele with >95% confidence (red line in Fig. 4f). In contrast to pairs of enhancer-enhancer and promoter-promoter bursts, pairs of enhancer–promoter bursts often occurred at distances <1 μm (Fig. 4f). Enhancer and promoter bursting on the same allele occurred significantly more frequently than expected

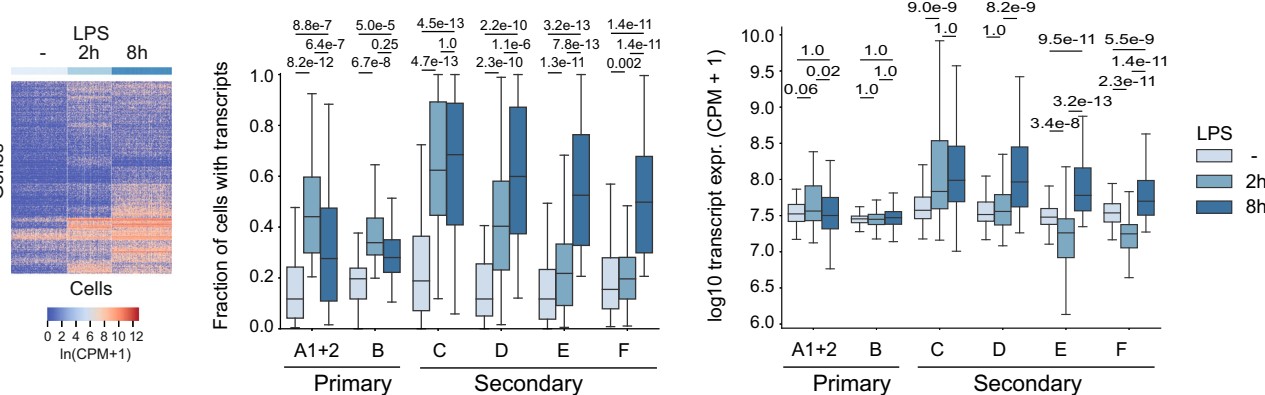

**Fig. 2 Inducible gene transcription is driven primarily by the fraction of expressing cells.** Left: heatmap of transcript levels (CPM) of inducible macrophage primary and secondary response genes[30] detected by scRNA-seq in at least 20% of cells. Centre: Fraction of cells with detectable transcripts for each of six previously defined classes of inducible genes[30] at 0, 2 and 8 h after LPS activation. Right: Transcript levels (CPM+1) per transcript-positive cell is shown for the same classes of inducible genes at 0, 2 and 8 h after LPS activation. Numbers represent adjusted *P* values for 0 versus 2 h and 0 versus 8 h. Two-sided Wilcoxon signed-rank test with Bonferroni correction for multiple testing. $N = 682$ cells.

by chance (Fig. 4g). Taken together, these data show that enhancer burst fractions are efficiently coupled to the burst fractions of target promoters, both at the cellular and the allelic level.

**Impaired expression of inducible genes in cohesin-deficient macrophages is the result of reduced transcription.** To examine the role of the chromatin-associated cohesin complex in the transcriptional bursting of enhancers and promoters, we deleted floxed *Rad21* (*Rad21^lox/lox^*) alleles encoding the cohesin subunit RAD21 by 4-OHT-mediated activation of ERt2Cre in bone marrow-derived mature macrophages[9]. This approach removed 90–99% of *Rad21^lox/lox^* alleles (Supplementary Fig. 5a) and ~90% of RAD21 protein (Supplementary Fig. 5b, c). Loss of RAD21 protein was uniform across individual cells, as demonstrated by immunofluorescence staining and confocal microscopy (Supplementary Fig. 5d, e). Mature macrophages are suitable for mechanistic studies on the role of cohesin in gene regulation because their quiescent state avoids interference with essential cell cycle functions of cohesin[39]. The conditional deletion of the cohesin subunit *Rad21* in quiescent primary macrophages results in a broad reduction in the expression of immune response genes as assessed by RNA-seq (ref. 9). Comparison of RNA-seq with GRO-seq showed that this reduction in inducible immune gene expression occurs predominantly at the transcriptional level in both resting and LPS-activated macrophages (Fig. 5a).

**Transcript copy numbers scale with burst fraction.** We used smRNA-FISH with exon probes to quantify the fraction of cells expressing inducible transcripts and the number of transcripts per cell, as exemplified for *Il12b* in Fig. 5b. *Rad21^−/−^* macrophages showed a reduced faction of transcript-expressing cells compared to wild-type macrophages (Fig. 5b). In contrast, the number of transcripts per cell was similar for *Rad21^−/−^* and wild-type macrophages (Fig. 5b). This indicates that the reduced expression of inducible transcripts in *Rad21^−/−^* macrophages is primarily due to a reduced frequency of cells expressing *Il12b* transcripts.

Consistent with these smRNA-FISH results, analysis by scRNA-seq showed that the frequency of individual cells expressing previously defined inducible transcripts[31] was reduced in *Rad21^−/−^* compared to wild-type macrophages, both at baseline and after LPS stimulation for 2 or 8 h (Fig. 5c). In contrast to the fraction of transcript-containing cells, the level of

transcripts per cell was similar for wild-type and *Rad21^−/−^* macrophages after 2 h of LPS stimulation.

To examine the transcriptional parameters associated with the reduced fraction of *Rad21^−/−^* macrophages expressing inducible genes, we used intron smRNA-FISH for *Egr2*, *Prdm1*, *Il12b*, *Ifnb1* and *Peli1*. In *Rad21^−/−^* macrophages, inducible genes displayed reduced burst fractions, while burst intensities were only mildly affected (Fig. 5d). These results suggest that the reduced expression of inducible immune genes in *Rad21^−/−^* macrophages at the population level was due primarily to a reduced probability of transcription, which led to a reduced fraction of cells expressing inducible immune genes.

**Transcriptional bursting is productive in the absence of cohesin.** To examine whether cohesin controls the number of mature transcripts, we compiled mRNA counts for cytoplasmic transcripts and burst parameters for the inducible genes *Cxcl10*, *Ifit1*, *Ifnb1* and *Il12b*, and compared the $\log_2$ burst measures against the $\log_2$ mRNA mean counts for each replicate in wild-type and in *Rad21^−/−^* macrophages. This analysis comprised a total of 77,482 cells from 76 samples, 38 for wild-type and 38 for *Rad21^−/−^* macrophages (Fig. 6a). Mature transcript copy numbers were highly correlated with burst fractions both in wild-type and *Rad21^−/−^* macrophages (Fig. 6a).

We next addressed whether wild-type and *Rad21^−/−^* macrophages produced the same number of *Cxcl10*, *Ifit1*, *Ifnb1* and *Il12b* transcripts for a given burst fraction over the range of burst fractions observed (Fig. 6b). For each gene, burst fraction and transcript copy number were highly correlated for both wild-type (black) and *Rad21^−/−^* macrophages (red) (Fig. 6b). There were no significant differences in the relationship between burst fraction and transcript copy number for any of the genes examined (Fig. 6b). This analysis showed that even though *Rad21^−/−^* macrophages showed reduced transcriptional burst fractions, they produced mature mRNAs in numbers proportional to the observed fraction of transcribing alleles. We further compared burst parameters observed in resting and LPS-activated wild-type and cohesin-deficient cells with transcriptional activity (GRO-seq, Supplementary Fig. 6a), and with the abundance of mature transcripts (RNA-seq, Supplementary Fig. 6b). Both transcriptional activity (GRO-seq) and the abundance of mature transcripts (RNA-seq) strongly correlated with the fraction of transcriptional bursting in both wild-type and *Rad21^−/−^*

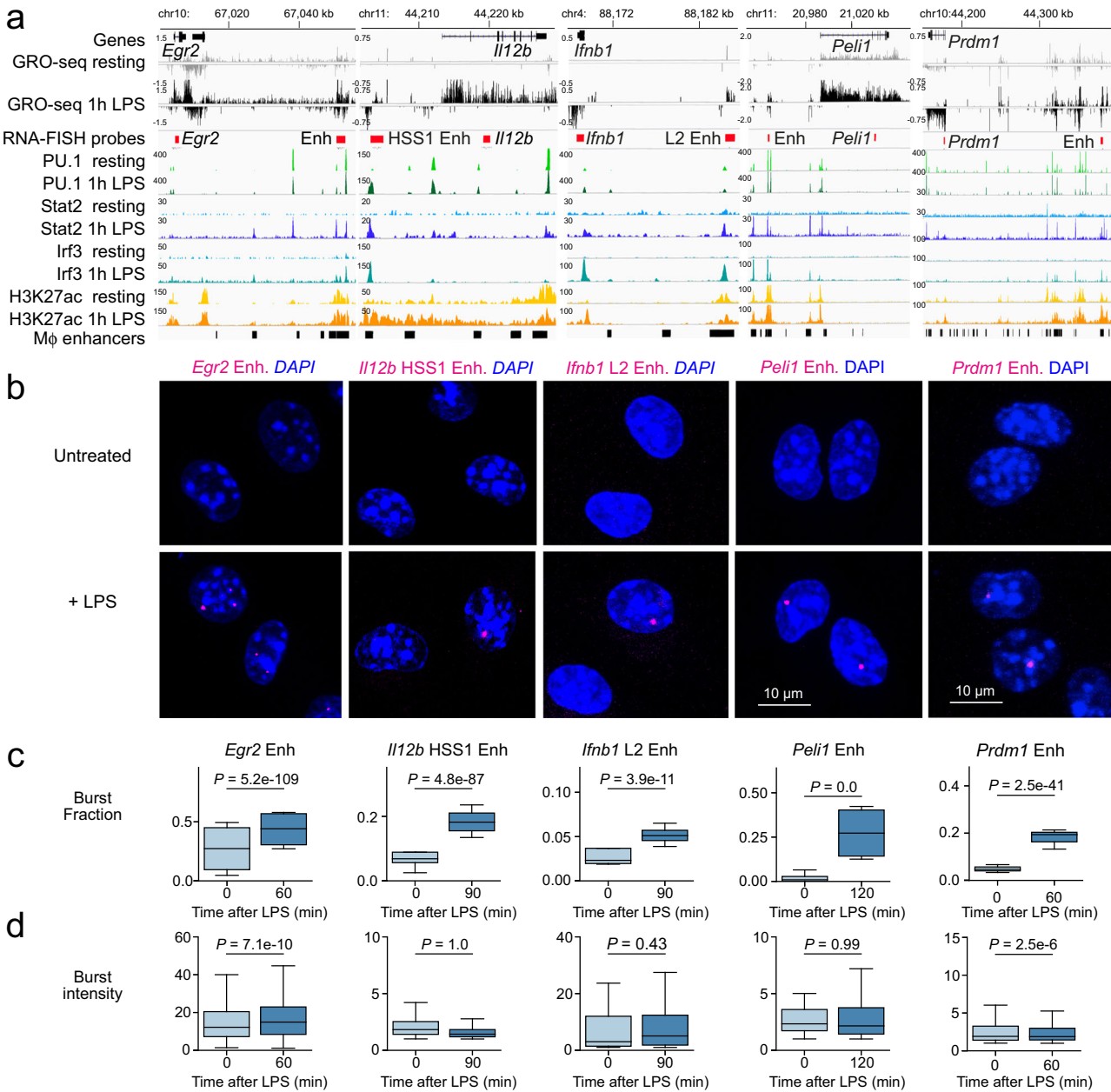

**Fig. 3 Inducible enhancer activation is linked to increased eRNA burst fractions. a** Inducible enhancers associated with inducible genes in macrophages with GRO-seq reads from resting and 1 h LPS-activated macrophages, ChIP-seq for PU.1 and STAT2, IRF3 and H3K27ac. Macrophage enhancers were defined previously[31]. The positions of smRNA-FISH probes are magenta. **b** smRNA-FISH Images of eRNA bursts in untreated and LPS-activated macrophages with probes for the *Egr2* enhancer (0 and 60 min LPS), the *Il12b* HSS1 and *Ifnb1* L2 (0 and 90 min LPS), and the *Peli1* enhancer (0 and 120 min LPS). smRNA-FISH probes are in magenta. At least three independent biological replicates per gene and condition, $n = 64{,}682$ cells. **c** eRNA burst fractions in resting and LPS-activated macrophages. At least three independent biological replicates per gene and condition, $n = 64682$ cells. **d** eRNA burst intensities in resting and LPS-activated macrophages. *P* values were determined by a two-sided Tukey HSD test (ANOVA design formula ∼ Sample + Replicate) and adjusted for multiple testing. At least three independent biological replicates per gene and condition, $n = 64{,}682$ cells.

macrophages (Supplementary Fig. 6). Taken together, these data indicate that cohesin is required for the initiation of transcriptional bursting, but, once initiated, the number of transcripts produced is a reflection of burst fraction, both in the presence and the absence of cohesin.

**Cohesin couples transcriptional burst fractions of inducible enhancers and their target genes.** A subset of inducible enhancers characterised by the binding of *Irf-* and *Stat-* family

transcription factors are dysfunctional in cohesin-deficient macrophages[9]. This is illustrated here for the *Ifnb1* L2 enhancer (Fig. 7a). However, of 1112 intergenic enhancers[31] that were detectably transcribed by GRO-seq and showed significant induction after 1 h of LPS treatment in wild-type macrophages, the great majority (1048 of 1112 or 94.2%) were transcribed to at least wild-type levels in *Rad21*[−/−] macrophages (adj. $P > 0.05$). Enhancers associated with *Il12b*, *Prdm1*, *Peli1* and *Egr2* were efficiently induced in *Rad21*[−/−] as well as wild-type macrophages

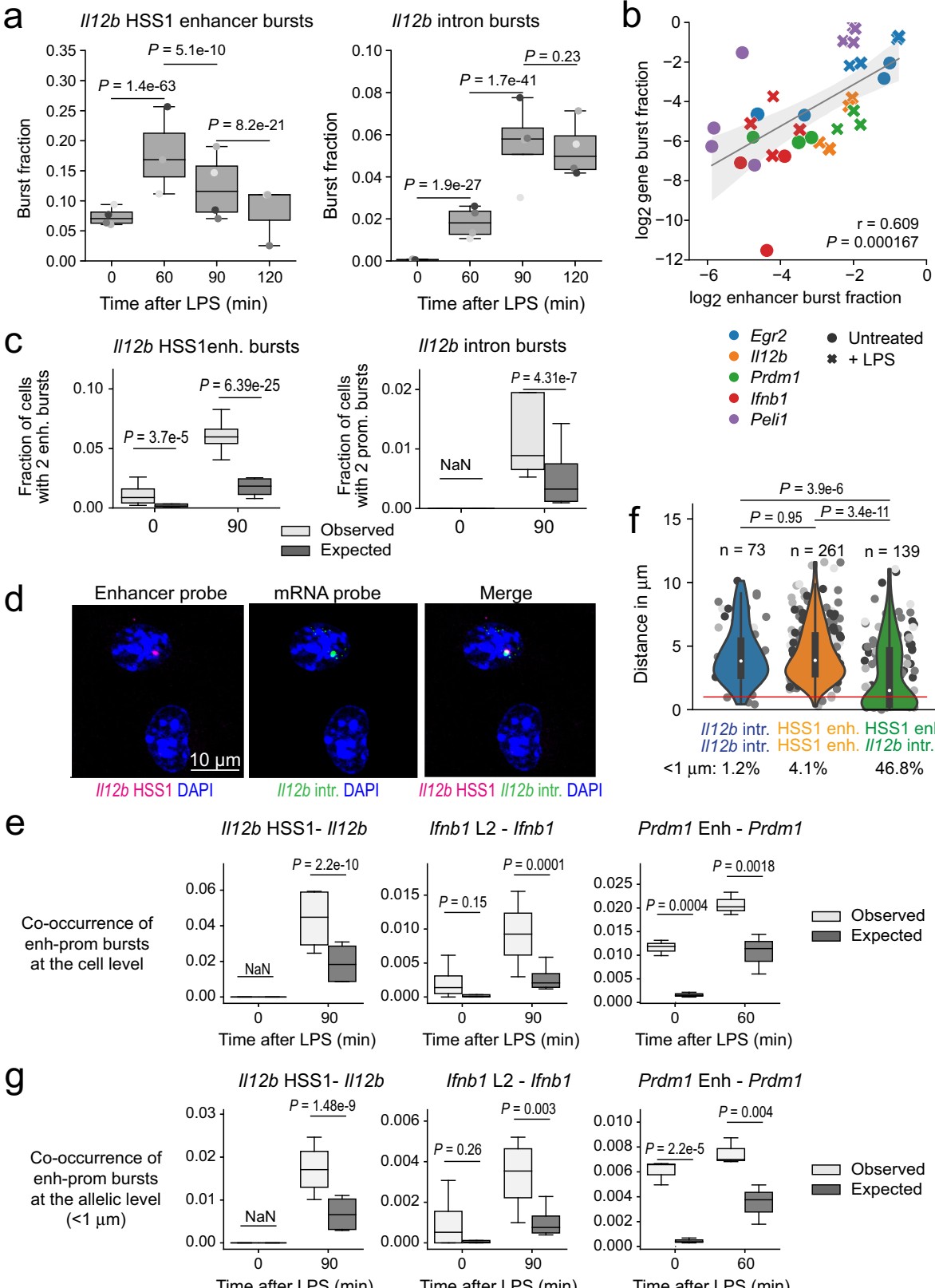

(Fig. 7a). The *Il12b* HSS1 enhancer, *Prdm1* enhancer and *Peli1* enhancer displayed higher burst fractions in *Rad21*⁻ᐟ⁻ compared to wild-type macrophages (Fig. 7a). The *Egr2* enhancer was significantly induced in *Rad21*⁻ᐟ⁻ macrophages, albeit to lower levels than in wild-type macrophages (Fig. 7a). In contrast to enhancer burst fractions, the promoter burst fractions of *Il12b*,

*Prdm1* and *Peli1* were markedly reduced in *Rad21*⁻ᐟ⁻ compared to wild-type macrophages (Fig. 7b). The promoter burst fraction of *Egr2* was more strongly reduced than the *Egr2* enhancer burst fraction in Rad21⁻ᐟ⁻ macrophages.

To quantify the observed defects in promoter activation we analysed the relationship between transcriptional burst fractions

**Fig. 4 Functional coupling of enhancer and promoter bursting. a** Transcriptional burst fractions of the *Il12b* HSS1 enhancer (left, 3–4 independent biological replicates, *n* = 14,759 cells) and *Il12b* (right, four independent biological replicates, *n* = 17,038 cells). **b** log$_2$-transformed burst fractions of the indicated enhancers and promoters in resting (UT) and LPS-activated macrophages (LPS). Two-way ANOVA formula (BF_Gene ~ BF_Enhancer*Gene). *P* value (P) for whether there is a relationship between the bursting of enhancers and genes (frequency_enhancer) = 1.94e-09. At least three independent biological replicates per gene and condition, *n* = 38093 cells. Error band: 95% confidence interval. **c** Bi-allelic enhancer (left) and promoter bursts (right) occur in individual cells more often than expected based on the burst fractions of enhancers and promoters (3–4 independent biological replicates, *n* = 14759 cells for *Il12b* HSS1 and *n* = 17038 cells for *Il12b*). Cochran–Mantel–Haenszel test (replicate as strata) with Bonferroni correction for multiple testing, two-sided considering a random population of the same size where the sampled frequency equals the expected frequency in case of bursting events happening independently. **d** Simultaneous application of enhancer (magenta) and target gene probes (green) illustrated for the Il12b locus. The position of probes is shown in Fig. 3a. More than eight biological and four technical replicates. **e** Experimentally observed (observed) co-occurrence of eRNA and mRNA transcriptional bursts over co-occurrence expected by chance (expected) at the level of individual cells. Cochran–Mantel–Haenszel test (replicate as strata) with Bonferroni correction for multiple testing, two-sided. At least three independent biological replicates per gene and condition, *n* = 38093 cells. **f** The distance distribution of *Il12b* intron - *Il12b* intron pairs, *Il12b* HSS1 enhancer—*Il12b* HSS1 enhancer pairs and *Il12b* intron—*Il12b* HSS1 enhancer pairs is plotted. A horizontal red line indicates a distance of 1 μm. The percentages of *Il12b* intron— *Il12b* intron pairs (1.2%), *Il12b* HSS1 enhancer—*Il12b* HSS1 enhancer pairs (4.1%) and *Il12b* intron—*Il12b* HSS1 enhancer pairs below 1 μm (46.8%) is indicated. Tukey HSD test (ANOVA design formula distance ~ pair + replicate), two-sided. **g** Box plots of experimentally observed (observed) co-occurrence of eRNA and mRNA transcriptional bursts over co-occurrence expected by chance (expected) at the level of individual alleles (i.e. at distances below 1 μm). Cochran–Mantel–Haenszel test (replicate as strata) with Bonferroni correction for multiple testing, two-sided. At least three independent biological replicates per gene and condition, *n* = 38,093 cells.

of genes and enhancers in wild-type and cohesin-deficient cells. In wild-type macrophages, enhancer activation was linked to efficient promoter activation (black in Fig. 7c), indicative of functional coupling between enhancers and promoters. *Rad21*$^{-/-}$ macrophages showed a blunted increase of target gene transcriptional burst fractions, despite elevated burst fractions of the *Il12b* HSS1, *Prdm1*, *Peli1* and *Egr2* enhancers in response to activation signals (red in Fig. 7c), indicating that enhancer–promoter coupling was substantially weakened (*Il12b*, *Peli1*, *Egr2*) or lost completely (*Prdm1*) in the absence of cohesin (Fig. 7d).

We next asked whether cohesin has a role in coupling the activity of inducible enhancers to target gene transcription genome-wide. To this end, we analysed the LPS-induced transcription of canonical macrophage enhancers[31] by GRO-seq (enhancer log2 fold-change, Fig. 7e) and the expression of protein-coding genes (RNA-seq, gene log2 fold-change, Fig. 7e) that were associated by 3D genome organisation at the level of TADs or contact domains. Wild-type macrophages showed a correlation between the transcription of enhancers and genes located in the same TADs (Fig. 7e left, r = 0.22 P = 4.09e-05) as well as contact domains (Fig. 7e right, r = 0.32, P = 5.67e-05). In contrast, as observed for the genes investigated by smRNA-FISH, cohesin-deficient macrophages showed a reduced correlation between the transcription of enhancers and genes located in the same TADs (Fig. 7e left, r = 0.12, P = 0.028) or contact domains (Fig. 7e right, r = 0.14, P = 0.01). The difference in the correlation between enhancer and gene transcription between wild-type and cohesin-deficient macrophages was significant at the level of both TADs (Fig. 7e left, P = 0.0006) and contact domains (Fig. 7e right, P = 1.99e-05). This analysis indicates that cohesin couples the activity of inducible macrophage enhancers to inducible macrophage genes genome-wide. Taken together with the smRNA-FISH data, these results show that the loss of cohesin uncouples transcriptional burst fractions of inducible genes from transcriptional burst fractions of inducible enhancers in innate immune cells and that enhancer-mediated gene activation involves discrete steps characterised by selective reliance on cohesin.

**Transcriptional bursts occur across a range of enhancer–promoter distances.** A straightforward interpretation of these results would be that cohesin mediates enhancer–promoter coupling simply by decreasing the proximity between enhancers and promoters in 3D nuclear space. Although appealing, the view of cohesin as a mere provider of spatial proximity faces significant

challenges: direct enhancer–promoter looping is neither strictly required, nor is it sufficient for the activation of target genes[40–44], and many loci that undergo enhancer–promoter looping during active transcription can do so in the absence of cohesin across significant genomic distances of ~100 kb (refs. [10,45–49]). Based on these and other considerations discussed more fully below, we set out to test the validity of the spatial proximity model of cohesin action in our experimental system.

Analysis of smRNA-FISH data showed that enhancer and promoter bursts occurred over a wide range of distances (Fig. 8, top). Even in the sub-micron range that contains enhancers and promoters on the same allele (see above), cumulative enhancer–promoter distances showed no—or at most minor—detectable differences between wild-type and cohesin-deficient cells (Fig. 8, bottom). The *Egr2* enhancer was slightly closer to the *Egr2* target gene in cohesin-deficient macrophages (genomic distance ~61 kb), and vice versa for *Peli1* (genomic distance ~43 kb). No significant differences were found for the *Prdm1* and *Il12b* enhancer-gene pairs (~198 and ~10 kb, respectively). These data show that transcriptional bursts can occur across a range of different enhancer–promoter distances. Furthermore, within the resolution limits of our experimental setup, enhancer–promoter distances are similar in wild-type and cohesin-deficient macrophages, suggesting that enhancer–promoter distances alone may not explain the role of cohesin in the functional coupling of enhancers and promoters.

**Non-additive effects of cohesin and BD2 perturbations.** We considered a potential involvement of BET proteins, since (i) human mutations in cohesin genes and in BRD4 cause related pathology[50–52], and (ii) either BET inhibition or targeted degradation of BRD4 abrogate transcription but leave enhancer–promoter contacts intact[53]. We used the BD1-selective small molecule inhibitor GSK778, which largely phenocopies pan-BET inhibitors, as well as the BD2-selective inhibitor GSK046, which has more limited effects on steady-state transcription[54]. BD1 and BD2 inhibitors primarily affected burst fractions rather than burst intensities (Fig. 9a). Selective inhibition of either BD1 or BD2 significantly blunted the LPS-induced increase of *Egr2* enhancer and *Egr2* promoter burst fractions in wild-type macrophages (Fig. 9a). Interestingly, selective inhibition of BD2 allowed for full LPS induction of burst fractions at the *Il12b* HSS1 enhancer, but significantly inhibited *Il12b* promoter burst fractions (Fig. 9a). Selective inhibition of BD2 therefore in effect uncoupled the *Il12b* HSS1 enhancer from the *Il12b* promoter, reminiscent of the uncoupling of enhancer and promoter bursting

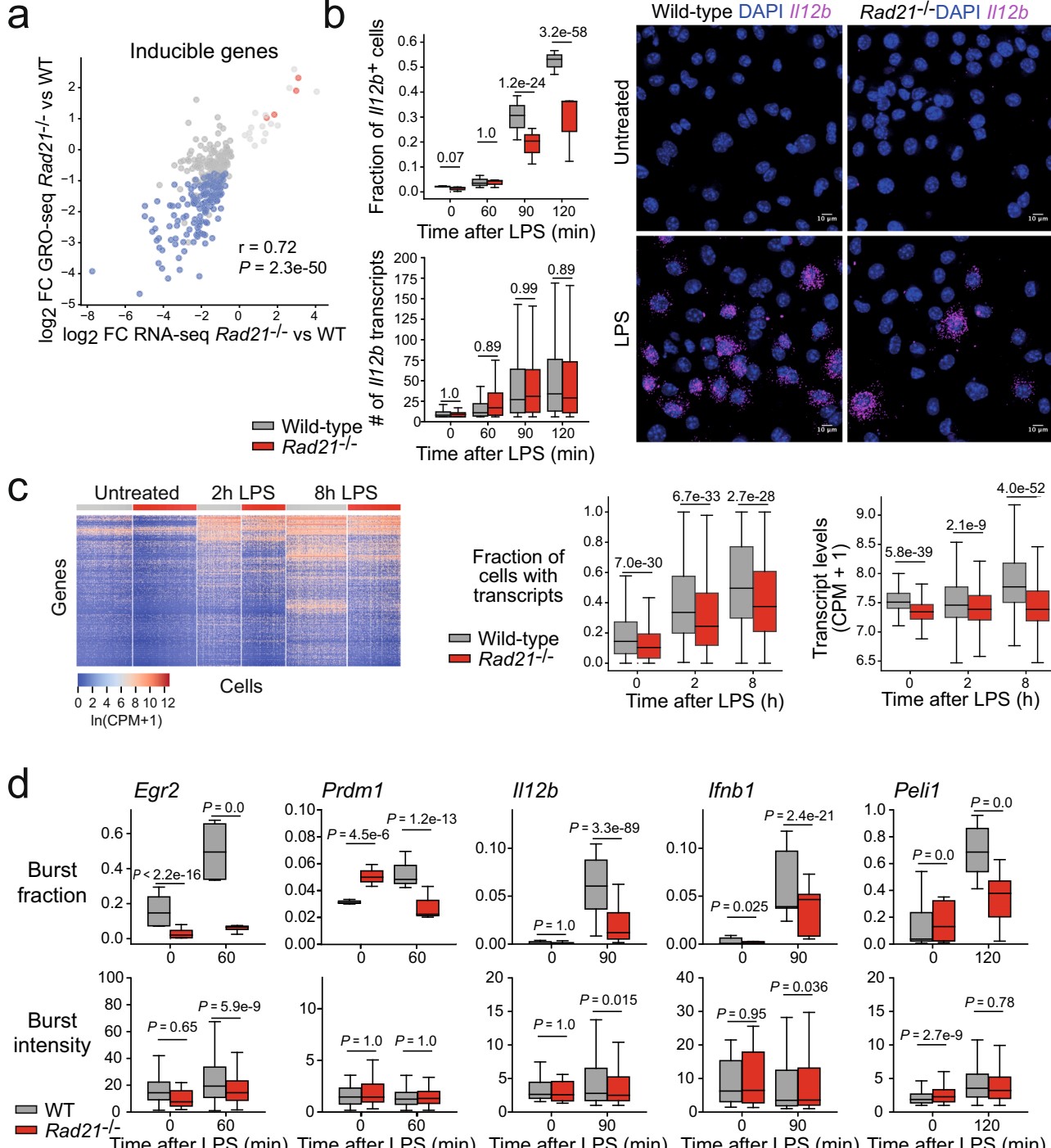

**Fig. 5 Deregulation of inducible genes in the absence of cohesin is due primarily to a reduced probability of transcription. a** Loss of cohesin alters the expression of inducible genes at the transcriptional level. One hour LPS for GRO-seq, 2 h LPS for RNA-seq. r Pearson correlation. Two-sided Pearson's product-moment correlation (null hypothesis: correlation = 0). Three biological replicates for RNA-seq and two biological replicates for GRO-seq. **b** The fraction of cells expressing *Il12b* by smRNA-FISH (top left) and the number of transcripts per cell (bottom left) are plotted for cells expressing at least five transcripts at the indicated time points after LPS activation of wild-type and *Rad21*$^{-/-}$ macrophages. Numbers represent adjusted *P* values. Wilcoxon signed-rank test with Bonferroni correction for multiple testing. Example images are shown on the right. N = 40,941 cells, four independent biological replicates. **c** Left: heatmap of inducible macrophage genes[30] detected by scRNA-seq in at least 20% of wild-type (grey) or *Rad21*$^{-/-}$ macrophages (red). Middle: Fraction of cells with detectable transcripts for inducible genes. Right: The expression level of LPS-inducible transcripts in cells with detectable transcripts is plotted as ln(CPM +1). Numbers represent adjusted *P* values. Two-sided Wilcoxon signed-rank test with Bonferroni correction for multiple testing. N = 1362 cells. **d** Burst fraction and burst intensity measurements for inducible immune genes in wild-type (black) and *Rad21*$^{-/-}$ macrophages (red). N = 11,8058 cells, 3–7 independent biological replicates.

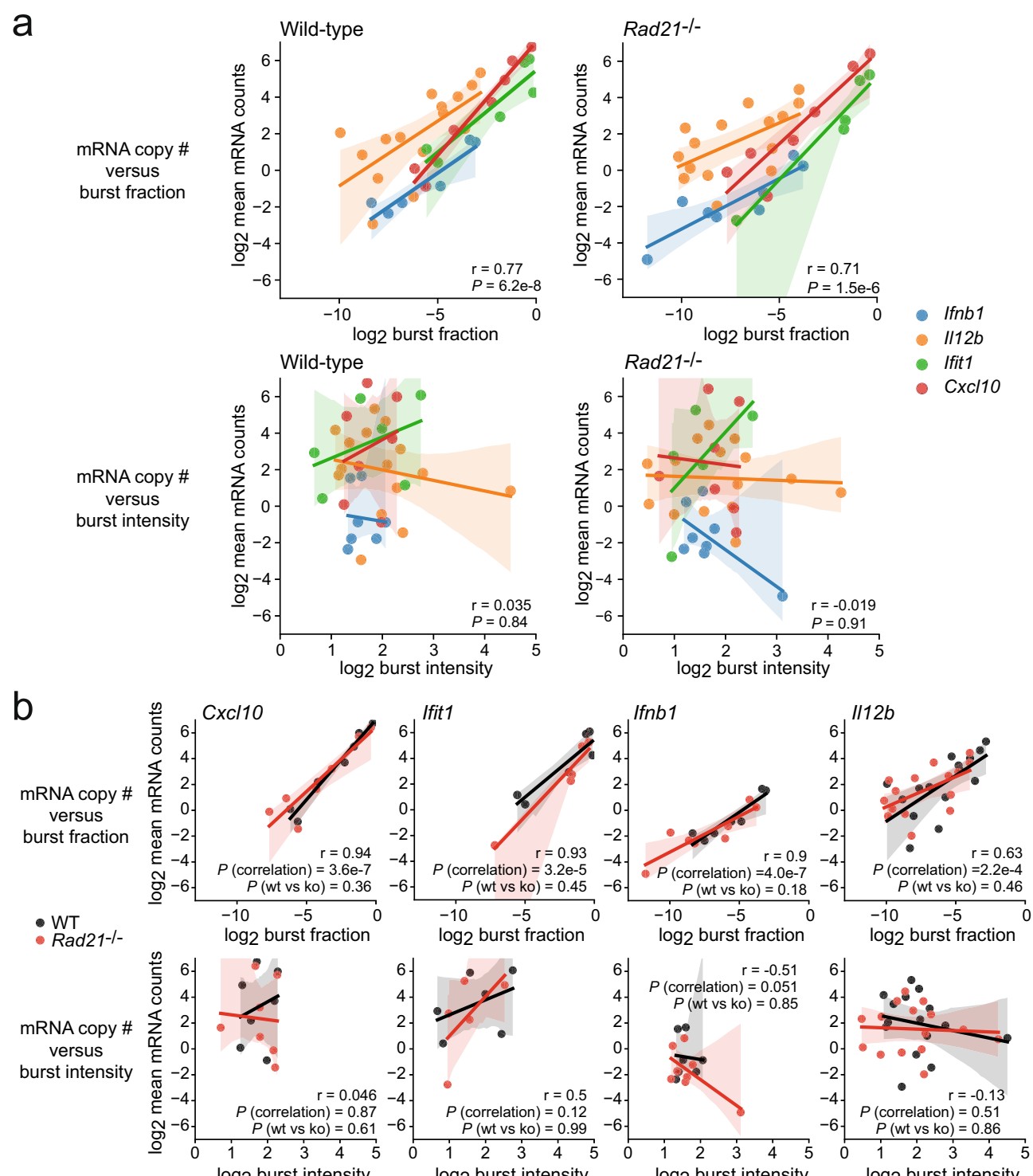

**Fig. 6 Transcriptional bursting is productive in the absence of cohesin. a** Transcript copy number correlates with transcriptional burst fraction in the absence of cohesin. Transcript copy number, a burst fraction (top) and burst intensity (bottom) were determined for wild-type (left) and $Rad21^{-/-}$ macrophages (right) as determined by smRNA-FISH using intronic probes for $Il12b$ and exon probes for $Ifit1$, $Cxcl10$, and $Ifnb1$. Shown are individual replicates for each gene in wild-type and $Rad21^{-/-}$ macrophages after 0, 30, 60, 90, 120 or 180 min of LPS stimulation. The overall correlation was tested. r Pearson correlation, error bands: 95% confidence interval. Two-sided Pearson's product-moment correlation (null hypothesis: correlation = 0). N = 78,747 cells, 3–4 independent biological replicates per transcript. **b** Transcript copy number, a burst fraction (BF, top) and burst intensity (BI, bottom) for wild-type (black) and $Rad21^{-/-}$ macrophages (red) as in Fig. 4d. For each gene we tested the overall r: Pearson correlation, error bands: 95% confidence interval. Two-sided Pearson's product-moment correlation against the null hypothesis: correlation = 0). Statistical test for differences between wild-type and $Rad21^{-/-}$ macrophages: two-way ANOVA in R with interaction formula l log2.mean.mRNA.counts ~ log2. Burst fraction * Genotype + log2. Burst intensity * Genotype. N = 78,747 cells, 3–4 independent biological replicates per transcript.

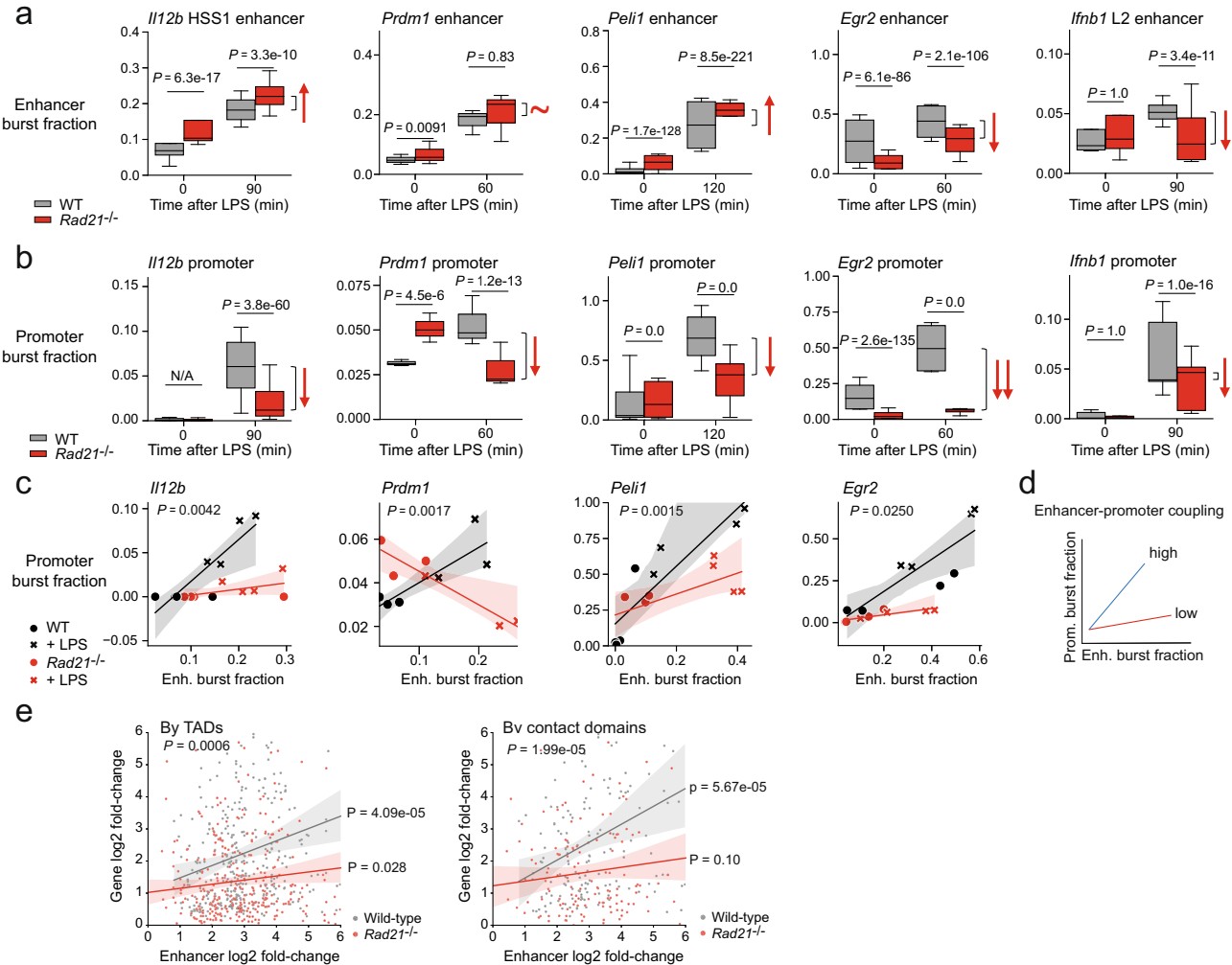

**Fig. 7 Cohesin couples transcriptional burst fractions of inducible genes and inducible enhancers. a** Transcriptional burst fractions of the indicated enhancers in wild-type (grey) and $Rad21^{-/-}$ macrophages (red). $N = 135,185$ cells, four independent biological replicates. **b** Transcriptional burst fractions of the indicated promoters in wild-type (grey) and $Rad21^{-/-}$ macrophages (red). $N = 135,185$ cells, four independent biological replicates. **c** Transcriptional burst fractions of the indicated promoters and enhancers in wild-type (grey) and $Rad21^{-/-}$ macrophages (red). Statistical test for the interaction: two-way ANOVA, two-sided (design formula: burst fraction gene ~ burst fraction enhancer*Genotype), $P$ values < 0.05 indicate that the relationship between enhancer and promoter burst fractions is different between $Rad21^{-/-}$ and wild-type macrophages. $N = 135,185$ cells, four independent biological replicates. Error bands: 95% confidence interval. **d** Schematic: Relationship between transcriptional burst fractions of inducible genes and inducible enhancers. **e** Correlation between the LPS-induced transcription of inducible enhancers (GRO-seq enhancer log2 fold-change 1 h after LPS) and the transcription of protein-coding genes located in the same TADs (left) or contact domains (right) as the enhancers (RNA-seq gene log2 fold-change 2 h after LPS) in wild-type (grey) and $Rad21^{-/-}$ macrophages (red). For TADs and contact domains with more than one enhancer, the mean fold-change is shown. TADs: Pearson correlation wild-type = 0.22, $P = 4.09e-05$. Pearson correlation $Rad21^{-/-} = 0.12$, $P = 0.028$. Two-way ANOVA Enhancer log2 fold-change:Genotype $P = 0.00061$. Contact domains: Pearson correlation wild-type = 0.32, $P = 5.67e-05$. Pearson correlation $Rad21^{-/-} = 0.14$, $P = 0.10$. Two-way ANOVA Enhancer log2 fold-change:Genotype $1.99e-05$. Error bands: 95% confidence interval.

observed in cohesin-deficient macrophages. To ask whether cohesin and BD2 might converge on a shared pathway we explored the impact of selective BD inhibitors on burst fractions in wild-type and cohesin-deficient macrophages. Selective BD1 inhibition reduced promoter burst fractions in wild-type and in both cohesin-deficient macrophages. Interestingly, however, selective inhibition of BD2 reduced *Egr2* and *Il12b* promoter burst fractions in wild-type, but not in cohesin-deficient macrophages (Fig. 9b). Hence, BD2 inhibition reduced *Egr2* and *Il12b* transcriptional burst fractions in the presence of cohesin, but had no additive effect to the loss of cohesin in $Rad21^{-/-}$ macrophages. This finding suggests the possibility that cohesin and BD2 may act in a shared pathway.

Cohesin is essential for genome integrity during the cell cycle[39], and DNA damage and repair have been proposed as one potential

pathway linking mutations in cohesin-related genes to a CdLS-associated Y430C mutation in the BD2 of BRD4 (Ref. [55]). In rapidly dividing mouse embryonic stem cells, corresponding BD2 mutations caused a delay in cell cycle progression, susceptibility to DNA damage, and delayed DNA damage repair[55]. We, therefore, examined cell cycle- and DNA damage-related pathways in wild-type and $Rad21^{-/-}$ macrophages.

Genes related to DNA damage, DNA repair and the cell cycle were downregulated in response to LPS activation in both wild-type and $Rad21^{-/-}$ macrophages (Supplementary Fig. 7a). Other than that, $Rad21^{-/-}$ macrophages showed no significant enrichment of gene ontology terms related to DNA damage, DNA repair or the cell cycle in either resting or LPS-activated conditions (Supplementary Fig. 7a). Gene set enrichment analysis of Hallmarks related to DNA

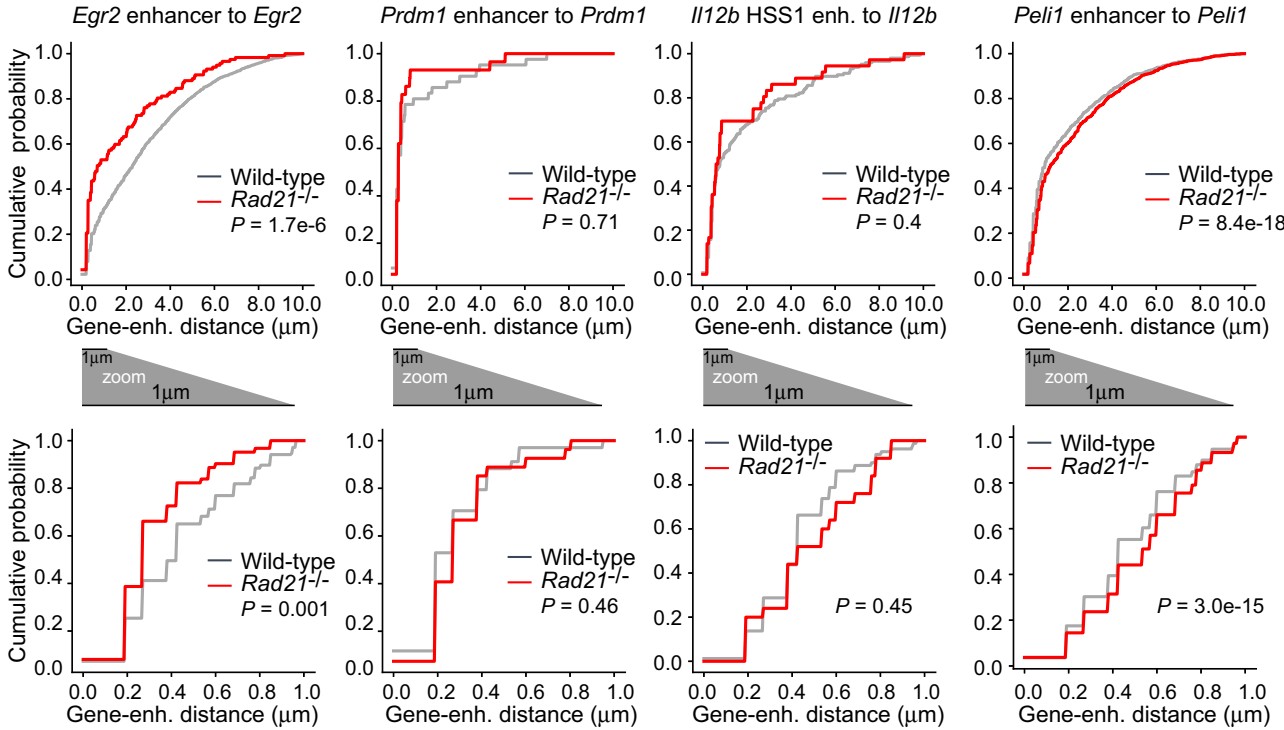

**Fig. 8 Similar enhancer–promoter distances in wild-type and cohesin-deficient macrophages.** Cumulative distances between enhancer and promoter smRNA-FISH signals in LPS-activated macrophages at the indicated loci. Number of wild-type cells: 8382 (*Egr2*), 3985 (*Prdm1*), 16,604 (*Il12b*), 19,864 (*Peli1*), at least three biological replicates per locus. Number of *Rad21*⁻/⁻ cells: 7658 (*Egr2*), 3992 (*Prdm1*), 7352 (*Il12b*), 18,600 for *Peli1*), at least three biological replicates per locus. Top: all observed distances. Number of wild-type gene-enhancer pairs: 1754 (*Egr2*), 59 (*Prdm1*), 370 (*Il12b*), 5004 (*Peli1*). Number of *Rad21*⁻/⁻ gene-enhancer pairs: 128 (*Egr2*), 35 (*Prdm1*), 43 (*Il12b*), 4288 (*Peli1*). Bottom: Distances up to 1 µm. Number of wild-type gene-enhancer pairs: 534 (*Egr2*), 23 (*Prdm1*), 116 (*Il12b*), 1440 (*Peli1*). Number of *Rad21*⁻/⁻ gene-enhancer pairs: 52 (*Egr2*), 20 (*Prdm1*), 19 (*Il12b*), 956 (*Peli1*). *P* values < 0.05 indicate that two cumulative distributions are different by the two-sided Kolmogorov–Smirnov test.

damage and the cell cycle[56] (MSigDB) revealed no major differences between wild-type versus *Rad21*⁻/⁻ macrophages (Supplementary Fig. 7b). *Rad21*⁻/⁻ macrophages showed reduced expression of G2M checkpoint and E2F targets compared to wild-type at 2 h after LPS activation, and marginally higher expression at late time points not used in our smRNA-FISH experiments. Genes that are downregulated during the response to UV damage were marginally upregulated in resting *Rad21*⁻/⁻ macrophages. Inflammatory pathways are known to be downregulated in *Rad21*⁻/⁻ macrophages and are shown as a control (Supplementary Fig. 7b). The absence of DNA damage or cell cycle effects is consistent with the quiescent state of mature macrophages (~90% G1) and the lack of p53 target gene activation in *Rad21*⁻/⁻ macrophages noted in our previous work[9]. Taken together, these results indicate that the pathways shared by cohesin loss and BD2 inhibition in macrophages are unrelated to DNA damage.

## Discussion

Here we dissect the regulation of burst parameters in inducible gene transcription and the cooperation between enhancers and their target genes. We show that inducible immune genes and enhancers in primary mouse macrophages are regulated predominantly by the fraction of transcriptional bursting, while burst intensities remain largely unchanged. We extend previous observations that inducible immune genes show heterogeneous expression at the single cell level by scRNA-seq (ref. [38]) by demonstrating that cohesin loss altered the frequency of cells that express inducible immune gene transcripts, rather than the level of expression per cell. Consistent with these results, smRNA-FISH showed that loss of cohesin predominantly reduced burst fractions, while burst intensities were largely unaffected.

Inducible immune gene expression during the LPS response of macrophages is transient and involves a chain of hierarchical events that alter mRNA distributions dynamically. Therefore, these distributions cannot be assumed to be in steady-state, especially at early time points[30]. During this transient response, the expression of many inducible immune genes is biphasic[38], indicating that only a subset of cells activates the expression of a given gene during any one activation cycle. In this setting, cell to cell variability likely reflects the probability of a gene turning on at least once during the activation cycle, and the smRNA-FISH burst fraction captures the probability of alleles initiating transcription. Despite the observed increase in enhancer burst fractions in response to activation, loss of cohesin reduced coupling of enhancer and target gene bursting. Accordingly, scRNA-seq data and smRNA-FISH counts in cohesin-deficient macrophages show a reduction in the frequency of transcript-expressing cells, rather than in transcript copy number per cell. These observations are reminiscent of models where enhancer function affects the probability rather than the level of gene expression[57] and support the notion that cohesin couples target gene transcription to enhancer activity.

A simple interpretation of these results would be that cohesin mediates enhancer–promoter coupling by increasing the proximity between enhancers and promoters in 3D nuclear space. Consistent with this model, enhancers are major determinants of promoter burst frequency[15,58,59], DNA looping is known to predominantly affect burst frequencies[58–60], and cohesin loss has previously been linked to altered promoter burst frequencies in HCT-116 cells[8]. While appealing, the interpretation of cohesin as a mere enhancer–promoter glue faces several challenges: First, while enhancer–promoter looping is sufficient to activate transcription of certain genes[57,61,62], direct enhancer–promoter

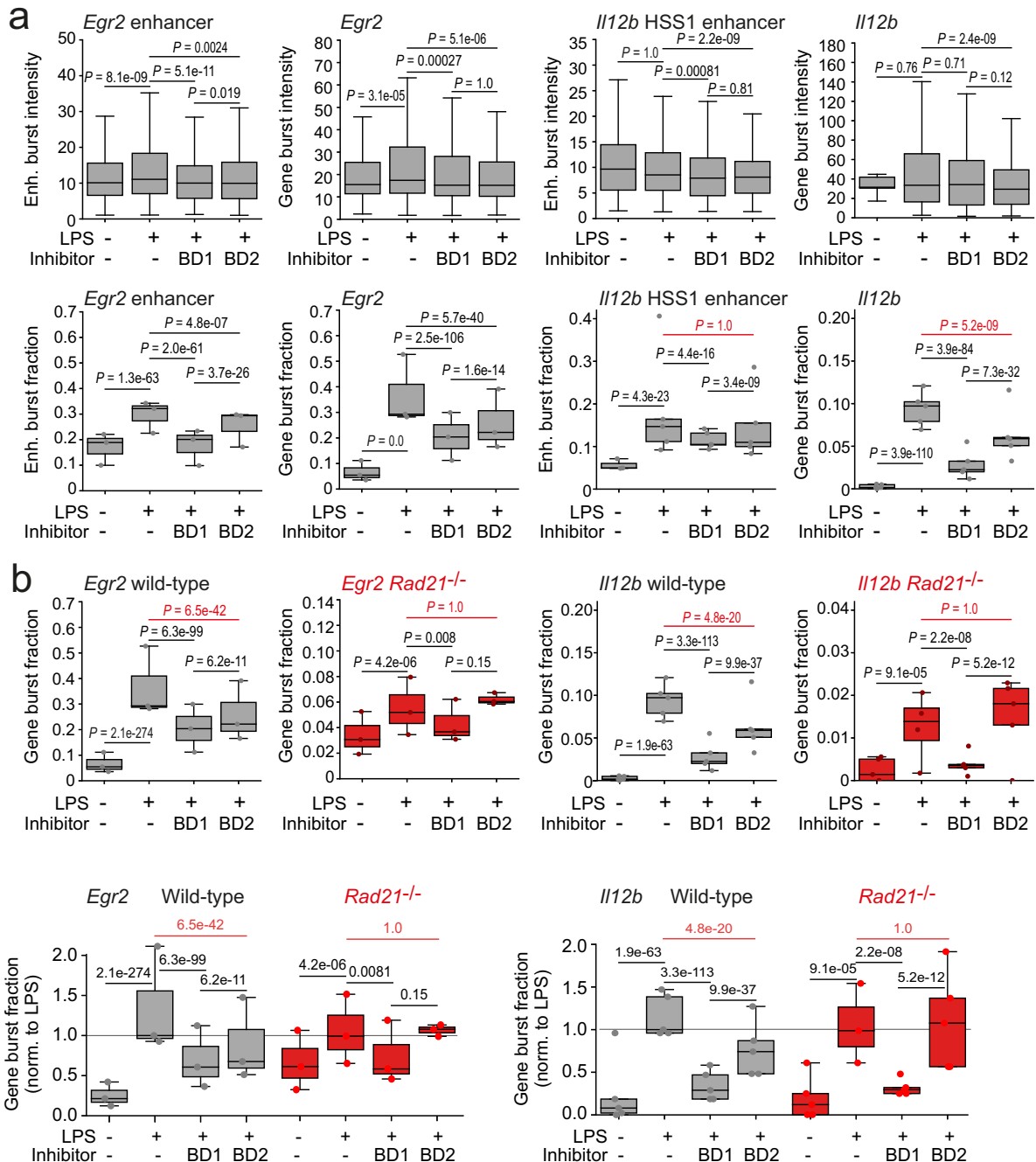

**Fig. 9 Perturbations of cohesin and the second BET bromodomain BD2 have non-additive effects on burst fractions. a** Transcriptional burst fractions of the *Egr2* enhancer, *Egr2*, the *Il12b* HSS enhancer and *Il12b* in wild-type macrophages 1 h after LPS activation in the presence of the BD1-selective inhibitor GSK778 (1μM), the BD2-selective inhibitor GSK046 (1 μM), or carrier (DMSO). *Egr2* enhancer *Egr2*: $n = 12,032$ cells, three independent biological replicates. *Il12b* HSS enhancer *Il12b*: $n = 17,610$ cells, 3–5 independent biological replicates. **b** Transcriptional burst fractions of *Egr2* and *Il12b* in wild-type (data as in **b**) and $Rad21^{-/-}$ macrophages 1 h after LPS activation in the presence of the BD1-selective inhibitor GSK778 (1μM), the BD2-selective inhibitor GSK046 (1 μM), or carrier (DMSO). *Egr2*: $n = 12,032$ wild-type and 9097 $Rad21^{-/-}$ cells, three independent biological replicates. *Il12b*: $n = 28,863$ wild-type and 23,155 $Rad21^{-/-}$ cells, 3–5 independent biological replicates.

looping appears to be neither strictly required nor sufficient for the activation of target genes[40–43]. For example, transcriptional bursting of *Sox2* appears uncorrelated with enhancer–promoter proximity[41,44]. Second, while active transcription of genes like *Shh*, *Fos*, and *Arc* is clearly associated with enhancer–promoter looping[10,45,46], contacts between enhancers and promoters at these[10,45] and other genes[47–49] occur robustly in the absence of cohesin across distances as far as ~100 kb. Third, promoter capture Hi-C in RAD21-depleted HeLa cells indicates that

contacts in the range of 10–100 kb are often retained, while contacts in the range of 100 kb and 1 Mb are mostly lost[63]. In agreement with these findings, enhancer and promoter bursts occurred over a range of distances in our experimental system. Within the resolution limits of our experimental system, cumulative enhancer–promoter distances showed little—if any—differences between wild-type and cohesin-deficient cells.

Selective inhibition of the BD2 bromodomain by GSK046 reduced gene transcription, but not enhancer activity at the *Il12b*

locus. This phenocopies the impact of cohesin depletion and implicates BD2 as well as cohesin in the functional coupling between enhancer activity and target gene transcription. Moreover, BD2 inhibition reduced *Egr2* and *Il12b* transcriptional burst fractions in wild-type macrophages but had no additive effect to the loss of cohesin in RAD21-deficient macrophages. Taken together with the observation that inducible genes appear to be particularly dependent on both BD2 (ref. [54]) and cohesin[9,10], these findings point to similarities between cohesin and BD2. To complement the inhibitor data presented here, future experiments will address the impact of BD2 mutations on transcriptional burst parameters.

Human CdLS is mainly associated with mutations in cohesin-related genes[50]. Strikingly, a missense mutation in the BD2 bromodomain of BRD4 causes CdLS-like developmental abnormalities[51]. Cell cycle delay, susceptibility to DNA damage, and defective DNA repair have been proposed as shared pathways triggered by mutations in cohesin-related genes and in BD2 (Ref. [55]). Our analysis shows that in quiescent macrophages, loss of cohesin does not trigger cell cycle- or DNA damage-related pathways. Our results do not question the observation that mutations in cohesin-related genes and in BD2 can lead to cell cycle aberrations and DNA damage in rapidly proliferating cells[55]. They do, however, show that neither cell cycle aberrations nor DNA damage are required for the shared effects of cohesin and BD2 on transcription.

BET proteins can impact transcription by a variety of mechanisms, which include RNA polymerase pause release and transcriptional elongation[64–66], as well as modulation of eRNA transcription and enhancer activity[32,66–68]. The availability of selective BD inhibitors and the identification of BD2 mutations in human disease[51] provide a path for future studies to link specific BET functions to specific protein domains. Such studies will be motivated not only by the desire to dissect the regulation of transcription but also by the relevance of BET proteins to human development[51,52] and cancer[54,64,65]. In this regard, mapping the mechanisms that functionally link BET proteins and cohesin is important for understanding the selective advantage of somatic cohesin mutations in haematopoietic malignancies and solid tumours[69]. In summary, our results delineate cohesin-dependent and -independent steps in enhancer-driven gene activation. Cohesin functionally couples the burst frequencies of inducible enhancers and promoters by mechanisms beyond spatial proximity.

## Methods

**Mice and cell culture**. Mouse work was performed according to the Animals (Scientific Procedures) Act Mouse work was done under a project licence issued by the UK Home Office, UK following review by the Imperial College London Animal Welfare and Ethical Review Body (AWERB). Bone marrow cells from *Rosa*26-ERt2Cre *Rad21*[WT/WT] or *Rad21*[lox/lox] mice[9] on a mixed C57BL/6 129 background were cultured in complete DMEM medium (10% FCS, 1% Penicillin-Streptomycin, 0.05 mM β-mercaptoethanol, 2 mM L-glutamine, 1 mM Na Pyruvate), 20% L929-conditioned media. Cre was induced on day 4 by 200 nM 4-hydroxy tamoxifen (Sigma-Aldrich H7904). Macrophages were stimulated on day 7 with 10 ng/ml of LPS from *Salmonella typhosa* (Sigma-Aldrich L7895). Where indicated, 1 μm GSK778 or GSK046 or carrier (DMSO) were added at the same time as LPS.

**Immunoblots**. Cell lysates were separated by SDS-PAGE on 7.5% gels (100 V, 90 min) and transferred to nitrocellulose membranes (90 V, 90 min). Membranes were blocked with 5% milk in Tris-buffered saline (TBS) with 0.1% Tween-20 and incubated with primary antibodies against RAD21 (Abcam 154769, 1/1000) and Tubulin (Sigma T9026, 1/1000) in TBS 3% milk overnight at 4 °C, washed 3x in TBS, incubated with Alexa fluor 680 goat anti-rabbit (Life Technologies A21109 1/10.000) to detect RAD21 or Alexa fluor 680 goat anti-mouse (Life Technologies A21057, 1/10.000) to detect Tubulin for 1 h at 4 °C, washed 3x in TBS, imaged (LI-COR Biosciences) and analysed (Image Studio Lite, LI-COR Biosciences).

**Immunofluorescence staining and confocal microscopy**. Sterile 13 mm coverslips were seeded with $2 \times 10^5$ cells and kept in a cell culture incubator at 37 °C overnight. Coverslips were washed with PBS and fixed for 5 min with 300 μL ice-cold 100% methanol at room temperature followed by three washes with PBS and stored at 4 °C overnight. Cells were permeabilised with 200 μL/sample 0.5% Triton X-100 in PBS for 10 min, incubated with 10% serum for 30 min, and primary antibody incubation was performed at 4 °C overnight using RAD21 (Abcam ab154769) 1:500 in 10% serum. Cells were washed three times with PBS and secondary antibody incubation was performed with goat α-rabbit 488 (Invitrogen A11034 1:500) for 1 h at room temperature. Confocal stacks were acquired in Leica SP8 microscope with a voxel size of $0.144 \times 0.144 \times 0.99$ μm, ×40 oil objective. For the analysis, nuclei and cell outlines were identified from maximum projections by CellProfiler v2.2.0 (ref. [70]) as primary and secondary objects. To normalise variable levels of background, RAD21 expression for each cell was defined as the mean nuclear intensity of the antibody channel minus the cytoplasmic mean. Example images were filtered to remove speckle background noise using a CellProfiler filter.

**smRNA-FISH**. Cells were seeded at a density of $2 \times 10^5$ per 13 mm coverslips or $1 \times 10^5$ per well in eight well chamber slides (Ibidi, 80826), kept in a cell culture incubator at 37 °C overnight, and activated with LPS where indicated. In experiments using exon probes, cells were stained with CellTracker (Invitrogen C34552 or C2925, 5 nM, 15 min, 37 °C) to facilitate the assignment of cytoplasmic smRNA-FISH signals to individual cells. Locus-specific exon, intron or enhancer smRNA-FISH probes (Supplementary Table 1) and View RNA ISH cell assay kits (QVC0001) were purchased from Affymetrix and used as advised by the manufacturer. To increase throughput, we used the QuantiGene View RNA HC Screening Assay (QVP0011, Affymetrix) protocol on chamber slides with the following modifications: First, volumes were doubled. PBS buffers contained 2 nM of an RNAse inhibitor (Ribonucleoside Vanadyl Complex, NEB S1402S). Cells were fixed in 4% formaldehyde + 2.5% of Glacial Acetic Acid to detect nuclear transcripts. Chamber slides were stained with DAPI (Thermo Scientific, 62248) 1:3000 in PBS for 5 min at room temperature. Coverslips were mounted using ProLong™ Gold Antifade with DAPI (Thermo Scientific, P36931). Samples were stored at 4 °C and imaged within 5 days.

**smRNA-FISH image acquisition and analysis**. Coverslips were imaged using a Leica TCS SP8 inverted microscope with an oil immersion 40×/1.30 NA. For each sample, 15 to 30 imaging fields were selected manually with a pixel size of 101.4 nm and a z-step size of 300 nm. Image analysis used the MATLAB version of Cell-Profiler (v1.0.0) with specific image-based transcriptomics modules[71,72] for nuclei, cytoplasm and dot identification.

To increase throughput, chamber slides were imaged with a Leica TCS SP5 inverted microscope with a dry 20 × /0.75 NA with a pixel size of 189 nm and a z-step size of 500 nm. The combination of a dry objective, chamber slides and MatrixScreener software allowed imaging of up to eight samples per night. The Leica TCS SP5 was used in resonant mode with a bidirectional scan, with a line average of 8–16 to reduce noise. Imaging fields were selected automatically, with human supervision only to ensure that cells were present. Fluorescence reflection was used for automatic focusing. Nuclei and cell outlines were identified from maximum projections using CellProfiler v2.2.0 modules as primary and secondary objects[70]. A size filter was applied to avoid scoring cell clusters as individual cells. Outlines were visually inspected, and then loaded in FISHQuant v3a (ref. [73]). FISHQuant counted mRNA molecules by fitting 3D Gaussian in the 3D stacks.

**Identification, quantification, and locus assignment of transcriptional bursts**. During a transcriptional burst, active TSSs accumulate signals that are brighter than individual transcripts. Burst intensities were estimated as the ratio between the integrated intensity of pixels at the TSS and the same area around averaged individual transcripts. Enhancers and TSSs were assigned to alleles based on distance (1 μm) using a custom Julia script (Julia v1.6, https://github.com/IreneRobles/TSSs). The expected co-occurrence of transcriptional bursts was calculated as the probability of one multiplied by the other. TSS-TSS distances were determined based on xy coordinates. Chromatic aberration was measured at centroid distances using 0.1 μm TetraSpeck™ microspheres and was 90.8 nm between 488 and 633 nm (*Ifnb1* - *Ifnb1* L2 enhancer), 102.3 nm between 561 and 488 nm (*Il12b* - *Il12b* HSS1 enhancer and *Peli1*-*Peli1* enhancer), and 41.4 nm between 561 and 633 nm (*Prdm1* - *Prdm1* enhancer). In each case, the chromatic aberration was smaller than the xy pixel size of 189 nm, and the data was not corrected for chromatic aberration.

**Burst size and burst frequency based on the moments of mRNA distribution**. To infer mean μ and variance $\sigma^2$, 50 cells were sampled from each distribution 1000 times. Burst size was calculated as $b_m = \sigma^2/\mu$ and burst frequency as $f_m = \mu/(b_m-1)$.

**RNA-seq analysis**. 100 bp paired-end RNASeq reads were aligned to mouse genome mm9 using Tophat2 (ref. [74]) with arguments '–*library-type fr-first strand -b2-very-sensitive -b2-L 25*' with gene annotation from Ensembl version 67. Read counts on genes were summarised using HTSeq-count (ref. [75]). The RUVseq Bioconductor package was applied with RUV k = 3 (1.18.0; ref. [76]) for ERCC spike-ins and replicate samples as controls. Differentially expressed genes were identified by DESeq2 (ref. [77]).

**GRO-seq analysis**. GRO-seq libraries were sequenced as 50 bp single-end reads in two biological replicates. The ten most 3′ bases were discarded based on fastqc quality assessment. Reads were aligned to mouse genome mm9 using bowtie with arguments '-l 30 -m 10 -n 2 –trim3 10'. Read counts on enhancers were computed using the summarizeOverlaps function from GeomicAlignments R Package. Differentially transcribed enhancers were identified using DESeq2.

**scRNA-seq**. WT and Rad21-deficient macrophages were sorted by flow cytometry into 386 well plates, and MARS-Seq libraries[78] were prepared and sequenced using two-level indexing of 192 cells per pool and multiple pools per sequencing lane. Sequencing was carried out as paired-end reads, wherein the first read contains the transcript sequence and the second read the cell barcode and UMIs. A quality check of the generated reads was performed with the FastQC quality control suite. Samples that reached the quality standards were then processed to deconvolute the reads to single-cell level by de-multiplexing according to the cell and pool barcodes. Reads were filtered to remove polyT sequences. Sequencing reads from human, mouse, or canine cells were mapped with the RNA pipeline of the GEMTools 1.7.0 suite[79] using default parameters (6% of mismatches, minimum of 80% matched bases, and minimum quality threshold of 26) and the genome references for human (Gencode release 24, assembly GRCh38.p5), mouse (Gencode release M8, assembly GRCm38.p4), and dog (Ensembl v84, assembly CanFam3.1). The analysis of spike-in control RNA content allowed us to identify empty wells and barcodes with more than 15% of reads mapping to spike-in artificial transcripts were discarded. Cells with less than 60% of reads mapping to the reference genome or more than $2 \times 10^6$ total reads were discarded. Gene quantification was performed using UMI-corrected transcript information to correct for amplification biases, collapsing read counts for reads mapping on a gene with the same UMI (allowing an edit distance up to two nucleotides in UMI comparisons). Only unambiguously mapped reads were considered. Genes not expressed in at least 5% of the cells were discarded.

**Statistical analysis**. Pearson correlation tests were done with the cor.test function in R. To test significant changes in event frequency across replicates we used a Cochran–Mantel–Haenszel test with replicate as strata variable in R. Bonferroni correction for multiple testing was implemented in MutipleTesting.jl Julia package. To test significant changes in distribution values across replicates, we used one-way ANOVA with Tukey HSD correction in R. To test whether the relationship between two variables differs between conditions we implemented two-way ANOVA in R.

**Reporting summary**. Further information on research design is available in the Nature Research Reporting Summary linked to this article.

## Data availability

scRNA-seq data generated for this study have been deposited at the Gene Expression Omnibus (GEO) under accession code GSE190622. smRNA-FISH images generated for this study have been deposited at the EMBL-EBI BioImage Archive under accession code S-BIAD338. The GRO-seq, RNA-seq, H3K27ac ChIP-seq data[9] used in this study are available in the GEO database under accession code GSE108599. Macrophage Hi-C data used in this study are available in the GEO database under accession code GSE115524. PU.1 ChIP-seq data[80] used in this study are available in the GEO database under accession code GSE56121. STAT2 ChIP-seq data[80] used in this study are available in the GEO database under accession code GSE56123. IRF3 ChIP-seq data[81] used in this study are available in the GEO database under accession code GSE67343. Source data are provided with this paper.

## Code availability

Code used in this study is available on Github, including code to make paper figures[82] https://github.com/IreneRobles/Code_Paper, TSS quantification and analysis from FISHquant outlines[83] https://github.com/IreneRobles/TSSs, processing of scRNA-seq data[84] https://github.com/IreneRobles/SingleCellExperiment, dimensionality reduction on matrix data[85] https://github.com/IreneRobles/DimensionalityReduction and scRNA-seq data[86] https://github.com/IreneRobles/SingleCellDimensionalityReduction, and to generate input files for FISHquant and CellProfiler from microscope lif files[87] https://github.com/IreneRobles/Make2Dand3Dstacks_fromlif.

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

## Acknowledgements

We thank Dr. Thomas A Milne (MRC Molecular Haematology Unit, Oxford) for the discussion and Dr. Vahid S Shahrezaei and Ilinca Patrascan for critical reading of the manuscript. This work was supported by the Medical Research Council UK and The Wellcome Trust (Investigator Award 099276/Z/12/Z to M.M.).

## Author contributions

I.R.-R., A.C.-S., S.W. and A.G.C. did experiments, I.R.-R., A.C.-S., E.M., S.W. and A.G.C. analysed data, I.R.-R. curated data, I.R.-R., M.M.K., C.W. and D.D. wrote scripts, H.H. devised the scRNA-seq pipeline, C.W. and D.D. devised the microscopy pipeline, I.R.-R., S.C., S.M., I.R., R.K.P., M.P.H.S., A.G.F. and M.M. conceptualised the study, IR-R and MM made figures and wrote the manuscript.

## Competing interests

The authors declare no competing interests.

**Additional information**

