## [Peer Review File · Nature Communications]

REVIEWER COMMENTS

Reviewer #1 (Remarks to the Author):

Authors address the role of cohesin and bromodomain in transcription, particularly how cohesin function to couple enhancers to promoters. They use macrophage cells to assay inducible transcription upon LPS treatment and measure nascent transcription by GROseq, single-cell transcriptomics and detect transcriptional burst size and frequency by smRNAFISH to dissect the effect of cohesin depletion on transcription burst frequencies and size of the burst. Authors convincingly show cohesin is essential for enhancer transcription burst frequency but not the burst size. Also, cohesin is important for coupling enhancer burst frequency to promoter burst frequencies. Finally, they link the role of cohesin in enhancer-promoter coupling using cohesinopathies associated mutation in bromodomain 2 of BET proteins by using BD2 specific drugs. Overall, this is an elegant and thorough study linking cohesin function to enhancer-promoter coupling and transcriptional bursts.

Conclusions and claims are supported by the data, and the methodology and stats well described.

However, only weaker part of the study is the proposed role of shared pathway between BD2 (using BD2 specific drugs) and cohesin depletion (Rad21^{-/-}).

Specific comments:

- In CdLS there are mutations throughout BRD4 including BD2, and also BD2 is not BRD4 specific inhibitor. Several evidence demonstrate interaction between BRD4 and cohesin for e.g. recent paper showing N-terminus of BRD4 (not BD2) shown to interact with NIPBL (PMID: 34611363). Authors should discuss the caveat of linking BD2 to cohesin pathways using BD2 specific drugs approach instead of using mutations in BRD4 domains.

Minor issues:

Page 19, second paragraph: it should be "simply by decreasing the proximity between enhancers and" instead of "increasing" and "enhances"

Figure legend 7: several times in this legend "number of gene enhancers" instead of "number gene-enhancer"

Page 25: paragraph 3: "They do" instead of "the do"

Page 25 paragraph 4 "Olley et al" instead of "Ollie et al"

Best wishes

Pradeepa Madapura

Reviewer #2 (Remarks to the Author):

In this paper, the authors use single molecule RNA FISH (smRNA-FISH) to measure transcriptional activity of inducible genes upon LPS mediated induction of macrophage differentiation. In a series of carefully calibrated experiments, they show that inducible gene expression is regulated by transcriptional burst frequency. Similarly, they showed that transcription at enhancers was also regulated by burst frequency upon LPS induction, and that enhancer activation preceded gene transcription. On a single cell level, they went on to show that enhancer burst frequency is coupled to the burst frequency of promoters on the same allele. Interestingly, they then found that in the absence of cohesin (via a Rad21 inducible deletion) the fraction of cells displaying transcriptional bursting activity was decreased (due to decreased burst frequencies), but the number of transcripts and

subsequent mature RNAs per cell stayed the same. This was associated with a maintained (or in some cases increased) enhancer burst frequency. They found that there are no changes in enhancer-promoter proximity with or without cohesin (as measured by smRNA-FISH), and that instead selective inhibition of BD2 indicates that BD2 and cohesin may function in the same pathway.

Overall, this is a carefully controlled elegantly performed study, where the data support the main conclusions. The insights from this paper are highly exciting and novel, and provide a strong contribution to the field. I only have two questions/minor concerns.

1. The main mechanistic issue with the paper is that it leaves the exact functional role of cohesin (and BD2) unknown, so the authors are presenting the lack of change in enhancer-promoter proximity as being one of the main conclusions of the paper. It is not clear to me if smRNA-FISH has the resolution needed to draw strong conclusions about enhancer-promoter proximity in the absence of cohesin. This is a minor point, but the authors need to acknowledge that it is possible that cohesin has subtle effects on enhancer-promoter proximity not resolvable by the imaging methods applied here. Either that, or maybe loss of cohesin is having an impact on promoter function in some unknown way that is disrupting transcription initiation. This is obviously an area for future extensive work that goes beyond the scope of this paper, but it would be good if some of these points were acknowledged.

2. Following from (1), a CTCF degron has minimal effects on gene expression genome wide, but strong effects on MYC expression and promoter-enhancer interactions (Hyle et al, MARE, 2019 PMID: 31127282). Presumably this is due to loss of cohesin at the distal enhancer, although this was not addressed directly in the paper. Is MYC regulation a special case, or is MYC CTCF-dependent regulation also compatible with the results here (in being independent of cohesin mediated looping)?

Reviewer #3 (Remarks to the Author):

The manuscript studies inducible gene expression programmes in innate immune cells using a combination of smRNA-FISH and scRNA-seq. They conclude that many genes are controlled by transcriptional bursting. They go on to study transcription using floxed Rad21 combined with ERT2Cre to knock down cohesin in mature macrophages. This results in a broad reduction in transcription in inducible immune genes. However, the number of transcripts per cell is maintained. They go on to show that knockdown of cohesin leads to uncoupling of transcription at the enhancers and promoters. Finally they show that BET inhibitors lead to effects on burst frequencies rather than burst sizes.

I find that this is a really interesting study, which addresses an important and timely question using a very elegant cellular model, which allows gene activation to be studied in primary cells. I think that the findings are supported by the data as presented and this study significantly advances our understanding of transcription. I am fully supportive of publication at this stage, however, I find that some of the conclusions are slightly overstated particularly in the abstract and I would like to see some additional technical detail.

Minor points:

'Burst frequency correlates with RNA-seq better than burst size'. This is an important point and I wondered how confident the authors were that burst size was being measured as accurately as frequency (the burst sizes seem to be low for several of the genes)? How

was an accurate value of the average single mRNA calculated? It would be helpful to know how many single mRNAs were measured to calculate the average intensity, and what the variability was. Also, what is the bit size of the images collected (this will dictate the intensity value range of the image data)? There seems to be considerable variability in the burst size and it would be useful if the authors could comment on what underlies this. The abstract states: "We show that inducible innate immune genes are regulated predominantly by the frequency of transcriptional bursting, and that burst frequencies are coordinated between enhancers and promoters.") I worry that this might be overstated?

'Burst fraction is related to the frequency of transcriptional bursting (Levesque and Raj, 2013, Padovan-Merhar et al., 2015), and is used here as a proxy for burst frequency.' It wasn't totally clear whether this is nascent or stable RNA transcripts (I assume nascent but it would be helpful to define burst frequency within this manuscript).

Fig1e. would it be possible to clarify what is plotted here. It is referring to an averaging for all the genes assessed by smFISH for the LPS treatment times stated?

Pg8 Fig2. For the enhancer burst sizes (burst size being defined in SupplFig1 as intensity of the transcription site/intensity of single average RNA). How was the average single eRNA intensity measured to make this calculation? The average value for the gene mRNA was calculated by identifying for single mature transcripts within the nucleus and measuring their intensities but this wouldn't be possible for the eRNAs?

Some enhancers are 'associated' with genes rather than being functionally validated. It would be useful to clarify how were they associated? In particular Supplemental Fig 6 could be strengthened to make the gene names visible and include publicly available Hi-C data (rather than TAD calls) / open chromatin / ChIP to confirm the position of the enhancers and their physical contacts with the genes.

'LPS increased the frequency of cells that expressed inducible transcripts for each of 6 previously identified inducible gene classes (A1/2, B, C, D, E, F; Bhatt et al., 2012) at both early and late time points (Fig. 1f, centre).' Brief explanation of the different classes would be helpful so that readers don't have to go to the original paper.

'In contrast to the fraction of transcript-containing cells, the level of transcripts per cell was similar for wild-type and Rad21^{-/-} macrophages after 2h of LPS stimulation were.' I think this sentence has a typo.

With regards to enhancer promoter contacts, I think that the authors should be clear that FISH for the RNA transcripts probably does not have sufficient resolution to directly visualise enhancer promoter contacts. In particular the Prdm and Il2b enhancer-gene distance data look remarkably similar despite distances of 198kb vs 10kb respectively. In addition, the z step sizes are reasonably large in this study. It would also be important to know if these data were corrected for chromatic aberration, which is inherent with any imaging. Combined with the reasonably large z step sizes this could lead to less accurate distance measures being made. Whilst I appreciate that there is debate about whether enhancers directly contact promoters in the literature, the most recent MNase based 3C approaches show extremely punctate contacts between enhancers and promoters, which are difficult to reconcile with models in which direct physical contacts are not required between enhancers and promoters. Although the authors have been careful in their wording in the discussion, I think the wording in the abstract is not supported by the data 'Enhancer-promoter coupling is not explained by spatial proximity alone'. I think the authors should add caveats and reduce the strength of the conclusions.

Finally the BET inhibitor findings are very interesting but again it is possibly over simplified in the abstract and discussion. Figure 8b is slightly confusing and might be helped if the Y axes were on the same scale. It appears that the BET inhibitors and Rad21 knockout have a different effect at Egr2 (where the eRNAs are reduced) compared to Il12b, where they don't change. It would be interesting if the authors could comment on this. It appears to be a complex phenomenon and the statements 'Cohesin is largely dispensable for the regulation of enhancer burst frequencies.' and 'Our data ... suggest that cohesin and BD2 act on a shared pathway.' are probably overstated.

Reviewer #4 (Remarks to the Author):

The paper by Robles-Rebollo focuses on the control of the inflammatory responses via transcriptional bursting using an impressive range and state-of-the-art single cell and population-level genomics and imaging approaches. The authors demonstrate that the control of the inducible gene expression in the TLR system is achieved by frequency modulation of transcription. Although this has been suggested before, the novel and important contribution of this paper is to link this transcriptional regulation to cohesin-mediated enhancer bursting. The most elegant part of the paper demonstrates how the enhancer bursting couples with promoter bursting (using a dual enhancer and intronic smRNA-FISH) and uses a mice model to demonstrate that cohesion regulates this coupling. The actual physical model of how cohesion achieves transcriptional control is not provided, but the authors use small molecules inhibitors to demonstrate that cohesion acts with BET proteins, but is unrelated to DNA damage.

Overall the manuscript includes an overwhelming amount of data, analyses are performed to a very high standard, while the figures and text are well written. I believe that the paper provides a step-change in our understanding of the regulation of gene expression in general and in the case of LPS-induced responses.

The authors use nascent smFISH-mRNA as the key readout for the burst size, and a fraction of actively transcribing alleles as bursting frequency. This methodology requires further validation. For example, the burst size, the quantity of mRNA produced in the single burst (kt/k_{off} in the random telegraph model), depends on the duration of the burst, not just on the transcription rate (as provided by the number of mRNA molecules at the transcription site, i.e.,). In turn, the fraction of actively transcribing alleles does not provide insight into the dynamical regulation, i.e., are we simply looking at one switch per allele (OFF->ON) upon activation, or indeed we have a transcriptional bursting process involving multiple switches of the same allele. Given that some data describes the very initial part of the response (up to 2h), which may indeed correspond to just a single switch, while other data extend to 8h of stimulation, this might change the context of the "bursting" described in the manuscript. The temporal aspect is also relevant for coupling between enhancer and promoter bursting, where the apparent differences in fractions could be due to underlying kinetics and different switching.

These issues can be addressed by comparison with the well-established estimates of burst size and frequency, using moments of the mRNA distribution (from smFISH and scRNA-seq) and using telegraph models to estimate transcriptional bursting parameters (which if needed may include information on nascent mRNA, see (Skinner et al. 2016)). Understanding the kinetic regulation of Il12b, mRNA promoter and enhancer (and other genes analyzed via smFISH) will clarify these issues.

A recent paper analyzed the role of burst frequency and size in the TLR-mediated responses

in macrophages, see (Bagnall et al. 2020), while others use similar modelling methodology to analyse control of NF- κ B transcription, e.g. (Bass et al. 2021). These should be acknowledged in the manuscript.

Other comments:

Fig. 1e legend is confusing, please clarify how normalisation was performed. Are matching RNA-seq time points available? Not clear from the description.

Fig. 1f. Most of the analyzed genes appear to have very low abundance, except of the first and last clusters (not clear if they correspond to A-to-F from the cited reference). Now authors use a different measure to demonstrate the dominance of burst frequency. The comparison with previously used measures should be shown, e.g. do burst size and frequency calculated using moments for this scRNA-seq data show the same trend, does smRNA-FISH data formatted as in f show this trend, do individual genes from b (e.g. I12b) show expected trends?

Fig. 2 b, burst size calculation relies on the size of the mature mRNA molecule, not clear from the images if they are produced. Please clarify.

Fig. 3. Enhancer bursting is more efficient than transcriptional bursting (~ 2 -fold higher difference in fraction of active cells, but lower burst size), suggesting that many enhancers bursts are not productive or not coordinated. These differences might be related to the underlying dynamics of the process, i.e. the frequency of multiple ON-OFF switching... This information can be extracted by analyzing smRNA-FISH data with telegraph models and should provide additional insight.

The presented data highlight the correlation between the enhancer and promoter bursting. However, does not indicate whether intron bursts occur without the corresponding enhancer bursting. This would be important to visualize for completeness. As above, does the absence of enhancer sequence completely abolish transcription from the locus, or just reduce the fraction/burst size of responding cells? While validation of I12b enhances has been achieved previously using population-level analyses, the single-cell interpretation remains unclear. Functional validation of enhancers would be an important addition to the manuscript, although this can be technically difficult.

Fig. 4. scRNA-seq data appears to be very sparse for the majority of the genes. It appears that only clusters presented at the top and bottom are robustly expressed (as in Fig 1f). Are we seeing a reduced number of drop-out events (due to sequencing coverage) as the expression increases in stimulated cells? Removing genes that are not robustly expressed should make this analysis more stringent.

Figs 7,8 and S7. Not convinced that the paper benefits greatly from the analyses in Fig. 8 (or S7). Perhaps should be moved to the supplementary info.

Bagnall, J., W. Rowe, N. Alachkar, J. Roberts, H. England, C. Clark, M. Platt, D. A. Jackson, M. Muldoon, and P. Paszek. 2020. 'Gene-Specific Linear Trends Constrain Transcriptional Variability of the Toll-like Receptor Signaling', *Cell Syst*, 11: 300-14 e8.

Bass, V. L., V. C. Wong, M. E. Bullock, S. Gaudet, and K. Miller-Jensen. 2021. 'TNF stimulation primarily modulates transcriptional burst size of NF- κ B-regulated genes', *Mol Syst Biol*, 17: e10127.

Skinner, S. O., H. Xu, S. Nagarkar-Jaiswal, P. R. Freire, T. P. Zwaka, and I. Golding. 2016. 'Single-cell analysis of transcription kinetics across the cell cycle', *Elife*, 5.

Response to REVIEWER COMMENTS NCOMMS-21-50303-T

Overall response

We thank the referees for their thoughtful and constructive comments

We have revised our manuscript to acknowledge the technical limitations of our study, in particular with respect to the resolution of our imaging methods, and the use of bromodomain inhibitors rather than bromodomain mutations.

We have also adjusted our narrative in places where the referees felt we had perhaps overstated our claims, in particular with regards to the convergence of cohesin- and BD2-dependent pathways.

We also acknowledge that our smRNA-FISH approach measures parameters that are distinct from burst sizes and burst frequencies as inferred mathematically from the moment of mature mRNA distributions. The revised manuscript describes the smRNA-FISH data in terms of the fraction of bursting alleles and the intensity of transcriptional bursts. To allow comparisons with previously reported parameters we have also added calculations of moment-derived burst frequencies and burst sizes.

A detailed point by point response follows below.

Reviewer #1 (Remarks to the Author):

Authors address the role of cohesin and bromodomain in transcription, particularly how cohesin function to couple enhancers to promoters. They use macrophage cells to assay inducible transcription upon LPS treatment and measure nascent transcription by GROseq, single-cell transcriptomics and detect transcriptional burst size and frequency by smRNAFISH to dissect the effect of cohesin depletion on transcription burst frequencies and size of the burst. Authors convincingly show cohesin is essential for enhancer transcription burst frequency but not the burst size. Also, cohesin is important for coupling enhancer burst frequency to promoter burst frequencies. Finally, they link the role of cohesin in enhancer-promoter coupling using cohesinopathies associated mutation in bromodomain 2 of BET proteins by using BD2 specific drugs. Overall, this is an elegant and thorough study linking cohesin function to enhancer-promoter coupling and transcriptional bursts.

Conclusions and claims are supported by the data, and the methodology and stats well described.

However, only weaker part of the study is the proposed role of shared pathway between BD2 (using BD2 specific drugs) and cohesin depletion (Rad21^{-/-}).

We thank the referee for the positive overall assessment of our study.

Specific comments:

- In CdLS there are mutations throughout BRD4 including BD2, and also BD2 is not BRD4 specific inhibitor. Several evidence demonstrate interaction between BRD4 and cohesin for e.g. recent paper showing N-terminus of BRD4 (not BD2) shown to interact with NIPBL (PMID: 34611363). Authors should discuss the caveat of linking BD2 to cohesin pathways using BD2 specific drugs approach instead of using mutations in BRD4 domains.

We agree with the referee and have added a statement to this effect to the discussion: 'We found no additive effect of BD2 inhibitors and loss of cohesin, suggesting that BD2 and cohesin may act in a shared pathway'

In addition, the revised discussion now clearly spells out that: 'To complement the inhibitor data presented here, future experiments will address the impact of BD2 mutations on transcriptional burst parameters'

Minor issues:

Page 19, second paragraph: it should be “simply by decreasing the proximity between enhancers and” instead of “increasing” and “enhances”

Figure legend 7: several times in this legend “number of gene enhancers” instead of “number gene-enhancer”

Page 25: paragraph 3: “They do” instead of “the do”

Page 25 paragraph 4 “Olley et al” instead of “Ollie et al”

We thank the referee for spotting these errors, which we have rectified in the revised manuscript

Reviewer #2 (Remarks to the Author):

In this paper, the authors use single molecule RNA FISH (smRNA-FISH) to measure transcriptional activity of inducible genes upon LPS mediated induction of macrophage differentiation. In a series of carefully calibrated experiments, they show that inducible gene expression is regulated by transcriptional burst frequency. Similarly, they showed that transcription at enhancers was also regulated by burst frequency upon LPS induction, and that enhancer activation preceded gene transcription. On a single cell level, they went on to show that enhancer burst frequency is coupled to the burst frequency of promoters on the same allele. Interestingly, they then found that in the absence of cohesin (via a Rad21 inducible deletion) the fraction of cells displaying transcriptional bursting activity was decreased (due to decreased burst frequencies), but the number of transcripts and subsequent mature RNAs per cell stayed the same. This was associated with a maintained (or in some cases increased) enhancer burst frequency. They found that there are no changes in enhancer-promoter proximity with or without cohesin (as measured by smRNA-FISH), and that instead selective inhibition of BD2 indicates that BD2 and cohesin may function in the same pathway.

Overall, this is a carefully controlled elegantly performed study, where the data support the main conclusions. The insights from this paper are highly exciting and novel, and provide a strong contribution to the field. I only have two questions/minor concerns.

We thank the referee for the positive overall assessment of our study.

1. The main mechanistic issue with the paper is that it leaves the exact functional role of cohesin (and BD2) unknown, so the authors are presenting the lack of change in enhancer-promoter proximity as being one of the main conclusions of the paper. It is not clear to me if smRNA-FISH has the resolution needed to draw strong conclusions about enhancer-promoter proximity in the absence of cohesin. This is a minor point, but the authors need to acknowledge that it is possible that cohesin has subtle effects on enhancer-promoter proximity not resolvable by the imaging methods applied here. Either that, or maybe loss of cohesin is having an impact on promoter function in some unknown way that is disrupting transcription initiation. This is obviously an area for future extensive work that goes beyond the scope of this paper, but it would be good if some of these points were acknowledged.

We agree with the referee that our imaging data do not rule out subtle effects of cohesin deficiency on enhancer-promoter proximity, which may not be not resolvable by our imaging methods. To acknowledge this possibility, the revised narrative now clearly states: "**Within the resolution limits of our experimental system, cumulative enhancer-promoter distances showed little - if any - differences between wild-type and cohesin-deficient cells**".

2. Following from (1), a CTCF degenon has minimal effects on gene expression genome wide, but strong effects on MYC expression and promoter-enhancer interactions (Hyle et al, MARE, 2019 PMID: 31127282). Presumably this is due to loss of cohesin at the distal enhancer, although this was not addressed directly in the paper. Is MYC regulation a special case, or is MYC CTCF-dependent regulation also compatible with the results here (in being independent of cohesin mediated looping)?

We thank the referee for this interesting question. The susceptibility of *MYC* expression to CTCF degradation may be due to the very large genomic distance between *MYC* and its + 1.8 Mb distal enhancer. In related studies, we (Calderon et al., 2021 bioRxiv 2021.02.24.432639) and others (Kane et al., 2021 Biorxiv 10.1101/2021.06.24.449812; Thiecke et al., Cell Rep. 2020 32: 107929; Rinzema et al., 2021 bioRxiv 10.1101/2021.10.05.463209) have found that the cohesin-dependency of 3D genomic contacts relates to the genomic distances traversed by chromatin loops, including loops between promoters and enhancers. Specifically, short loops can be maintained - or even form de novo (Calderon et al., 2021 bioRxiv 2021.02.24.432639) - in the absence of cohesin, whereas longer loops are more dependent on cohesin.

We hope that these changes go some way towards addressing the referee's concerns.

Reviewer #3 (Remarks to the Author):

The manuscript studies inducible gene expression programmes in innate immune cells using a combination of smRNA-FISH and scRNA-seq. They conclude that many genes are controlled by transcriptional bursting. They go on to study transcription using floxed Rad21 combined with ERT2Cre to knock down cohesin in mature macrophages. This results in a broad reduction in transcription in inducible immune genes. However, the number of transcripts per cell is maintained. They go on to show that knockdown of cohesin leads to uncoupling of transcription at the enhancers and promoters. Finally they show that BET inhibitors lead to effects on burst frequencies rather than burst sizes.

I find that this is a really interesting study, which addresses an important and timely question using a very elegant cellular model, which allows gene activation to be studied in primary cells. I think that the findings are supported by the data as presented and this study significantly advances our understanding of transcription. I am fully supportive of publication at this stage, however, I find that some of the conclusions are slightly overstated particularly in the abstract and I would like to see some additional technical detail.

We thank the referee for the positive overall assessment of our study.

Minor points:

The abstract states: “We show that inducible innate immune genes are regulated predominantly by the frequency of transcriptional bursting, and that burst frequencies are coordinated between enhancers and promoters.” I worry that this might be overstated?

We have revised the offending sentence abstract, which now reads: "We show that inducible innate immune genes are regulated predominantly by an increase in **the probability of active transcription**, and that **the probabilities of enhancer and promoter transcription are coordinated**". We hope this goes some way towards addressing the referee's concern.

‘Burst frequency correlates with RNA-seq better than burst size’. This is an important point and I wondered how confident the authors were that burst size was being measured as accurately as frequency (the burst sizes seem to be low for several of the genes)? How was an accurate value of the average single mRNA calculated? It would be helpful to know how many single mRNAs were measured to calculate the average intensity, and what the variability was. Also, what is the bit size of the images collected (this will dictate the intensity value range of the image data)? There seems to be considerable variability in the burst size and it would be useful if the authors could comment on what underlies this.

Average single mRNA intensities were calculated using the FISH-quant averaging function after batch processing and exclusion of outliers (e.g. high background, signal too high or too low) over all thresholded mRNAs. The typical number of mRNAs per replicate used for averaging was ~7000-8000. Referee 3 Fig. 1 illustrates the variability observed in the intensity of individual mRNAs for 3 replicates of one smRNA-FISH experiment. TSSs are

shown for comparison. Images are 8-bits and each pixel has a value between 0-255. To avoid confusion between burst sizes as inferred from the moment of mRNA distributions, and the intensity of signal at transcription start sites, we have now adjusted our nomenclature to describe the intensity of individual bursts as 'burst intensities'.

Referee 3 Figure 1. Variability observed in the intensity of individual mRNAs and of TSS observed by smRNA-FISH. Three of one smRNA-FISH experiment are shown. TSSs. Boxes show upper and lower quartiles and whiskers show 1.5 of the interquartile range. P-values were determined by two-sided T-test. Replicate 1: n = 1092 TSSs and n = 5688 single mRNAs. Replicate 2: n = 332 TSSs and n = 1573 single mRNAs. Replicate 3: n = 318 TSSs and n = 2531 single mRNAs.

'Burst fraction is related to the frequency of transcriptional bursting (Levesque and Raj, 2013, Padovan-Merhar et al., 2015), and is used here as a proxy for burst frequency.' It wasn't totally clear whether this is nascent or stable RNA transcripts (I assume nascent but it would be helpful to define burst frequency within this manuscript).

The referee raises an important point, which was also a concern for Referee 4. We have thoroughly revised our manuscript to acknowledge that our smRNA-FISH approach measures parameters that are distinct from burst size and burst frequency as derived mathematically from the moment of mature mRNA distribution. The revised manuscript describes the data in terms of the fraction of bursting alleles and the intensity of transcriptional bursts. To allow comparisons with previously reported parameters, the revised manuscript presents a direct comparison between allelic burst fractions as determined by smRNA-FISH with burst frequencies as derived from the moment of mature mRNAs, and also of burst intensities determined by smRNA-FISH with burst sizes as derived from the moment of mature mRNAs (New Fig 1e).

Fig1e. would it be possible to clarify what is plotted here. It is referring to an averaging for all the genes assessed by smFISH for the LPS treatment times stated?

Apologies for the muddled presentation in the original figure. We have now replaced Fig 1e with a comparative analysis of allelic burst fractions (as determined by smRNA-FISH) and burst frequencies (as derived from the moment of mature mRNAs). The new Fig. 1e also compares burst intensities (determined by smRNA-FISH) with burst sizes (as derived from the moment of mature mRNAs).

Pg8 Fig2. For the enhancer burst sizes (burst size being defined in SupplFig1 as intensity of the transcription site/intensity of single average RNA). How was the average single eRNA intensity measured to make this calculation? The average value for the gene mRNA was calculated by identifying for single mature transcripts within the nucleus and measuring their intensities but this wouldn't be possible for the eRNAs?

The revised manuscript now makes a clear distinction between 'burst size' as inferred from the moment of mRNA distributions, and 'burst intensity' as measured by smRNA-FISH. Enhancer burst intensities were determined as the brightness of eRNA probe signals at transcription start sites, and we have revised the narrative to clearly state this: "**Nascent smRNA-FISH captures the fraction of actively transcribing alleles and the intensity of probe signal at transcription start sites (Burst intensity, Fig.1c). We found that the inducible expression of *Ii12b* (Fig.1d) and other inducible genes tested by smRNA-FISH (*Egr2*, *Prdm1*, *Ifnb1*, *Peli1* and *Sertad2*, Supplementary Fig. 2) was associated primarily with an increase in the fraction of actively transcribing alleles. We compared the fraction of actively transcribing alleles, as directly quantified by smRNA-FISH, with the burst frequency as inferred mathematically from the moments (the mean and the variance) of mature mRNA distribution (Fig.1e, left; So et al., 2011; Raj et al., 2006; Bagnall et al 2000; Bass et al., 2021). We found good agreement, which indicates that burst fraction and burst frequency are correlated in LPS-activated macrophages. In contrast, burst intensity as quantified by smRNA-FISH poorly correlated with burst size as inferred by the moment of mature mRNA copy number distribution (Fig.1e, right)".**

Some enhancers are 'associated' with genes rather than being functionally validated. It would be useful to clarify how were they associated? In particular Supplemental Fig 6 could be strengthened to make the gene names visible and include publicly available Hi-C data (rather than TAD calls) / open chromatin / ChIP to confirm the position of the enhancers and their physical contacts with the genes.

We agree with this point and have revised Supplementary Fig 6 (now Supplementary Fig 3). Although we cannot determine enhancer-promoter loops due to the 1D proximity of enhancer and promoters at the loci studied and the resolution of the available Hi-C data, the new Supplementary Fig 3 shows contact domains as well as TADs, provides GRO-seq data for LPS-inducible enhancer transcription, and transcription factor binding at accessible chromatin sites previously defined as macrophage enhancers (Ostuni et al., 2013).

Supplementary Figure 3. Inducible enhancers associated with smRNA-FISH target genes.

We identified enhancers (Ostuni et al., 2013) that are both detectable by GRO-seq and show significant change between baseline and 1h LPS treatment of wild-type macrophages. This analysis identified 1112 inducibly transcribed enhancers in wild-type macrophages. Of these inducible enhancers, 1048 remained intact and 64 were downregulated (adj $P < 0.05$) in *Rad21*^{-/-} macrophages. Inducible enhancers located in the same TADs as smRNA-FISH target genes remain largely intact in cohesin-deficient macrophages. The Figure shows IGV screenshots of the TADs containing *Il12b*, *Egr2*, *Prdm1*, *Peli1* and *Ifnb1*. Intergenic macrophage enhancers are indicated (Ostuni et al., 2013). GRO-seq transcribed, LPS-inducible enhancers (adj. $P < 0.05$ at 1h LPS activation) are shown. Enhancers with reduced GRO-seq signal in *Rad21*^{-/-} macrophages are marked in red (adj. $P < 0.05$ at 1h LPS activation).

Along the same lines, and with the aim of presenting a higher resolution view of the 3D genome than that afforded by TADs, Fig. 6e of the revised manuscript now shows a genome-wide analysis of correlated enhancer transcription and gene expression at the level of contact domains. This analysis supports and extends our previous conclusion that the transcription of genes and enhancers is more coordinated in wild-type macrophages by demonstrating that the correlation between enhancer and gene transcription between is significantly higher in wild-type than in cohesin-deficient macrophages at the level of contact domains ($P = 1.99\text{e-}05$).

Figure 6e. Cohesin enables genome-wide correlation of enhancer transcription and gene expression at the level of TADs and contact domain

Correlation between the LPS-induced transcription of inducible enhancers (GRO-seq enhancer log2 fold-change 1h after LPS) and the transcription of protein-coding genes located in the same TADs (left) or contact domains (right) as the enhancers (RNA-seq gene log2 fold-change 2h after LPS) in wild-type (grey) and *Rad21*^{-/-} macrophages (red). For TADs and contact domains with more than one enhancer the mean fold-change is shown. TADs: Pearson correlation wild-type = 0.22, $P = 4.09\text{e-}05$. Pearson correlation *Rad21*^{-/-} = 0.12, $P = 0.028$. 2-way ANOVA Enhancer log2 fold-change:Genotype $P = 0.00061$. Contact domains: Pearson correlation wild-type = 0.32, $P = 5.67\text{e-}05$. Pearson correlation *Rad21*^{-/-} = 0.14, $P = 0.10$. 2-way ANOVA Enhancer log2 fold-change:Genotype $1.99\text{e-}05$.

‘LPS increased the frequency of cells that expressed inducible transcripts for each of 6 previously identified inducible gene classes (A1/2, B, C, D, E, F; Bhatt et al., 2012) at both early and late time points (Fig. 1f, centre).’ Brief explanation of the different classes would be helpful so that readers don’t have to go to the original paper.

We agree with the referee and have added the following sentence to the revised manuscript: "LPS increased the frequency of cells that expressed inducible transcripts for each of 6 previously identified classes of inducible primary (Fig. 1f, Bhatt classes A1/2, B; Bhatt et al.,

2012) and secondary response genes (Fig. 1f, Bhatt classes C, D, E, F; Bhatt et al., 2012). We have also added primary and secondary response genes to the display in Fig 1f)".

‘In contrast to the fraction of transcript-containing cells, the level of transcripts per cell was similar for wild-type and Rad21^{-/-} macrophages after 2h of LPS stimulation were.’ I think this sentence has a typo.

Thank you for pointing this out, we have fixed this typo.

With regards to enhancer promoter contacts, I think that the authors should be clear that FISH for the RNA transcripts probably does not have sufficient resolution to directly visualise enhancer promoter contacts. In particular the Prdm and Il2b enhancer-gene distance data look remarkably similar despite distances of 198kb vs 10kb respectively. In addition, the z step sizes are reasonably large in this study. It would also be important to know if these data were corrected for chromatic aberration, which is inherent with any imaging. Combined with the reasonably large z step sizes this could lead to less accurate distance measures being made. Whilst I appreciate that there is debate about whether enhancers directly contact promoters in the literature, the most recent MNase based 3C approaches show extremely punctate contacts between enhancers and promoters, which are difficult to reconcile with models in which direct physical contacts are not required between enhancers and promoters. Although the authors have been careful in their wording in the discussion, I think the wording in the abstract is not supported by the data ‘Enhancer-promoter coupling is not explained by spatial proximity alone’. I think the authors should add caveats and reduce the strength of the conclusions.

We agree with the referee's point and have revised the wording of the abstract to "Enhancer-promoter coupling **may not be** explained by spatial proximity alone". We have also revised the wording of the discussion to "**Within the resolution limits of our experimental system, cumulative enhancer-promoter distances showed little - if any - differences between wild-type and cohesin-deficient cells**".

It is true that the z step size in this study is too large to determine TSS-TSS distances with the required accuracy. For this reason the TSS-TSS distances shown are based on xy coordinates. The measured centroid distances of 0.1 µm TetraSpeck™ microspheres on our microscope are 90.8 nm between 488 and 633 nm (*Ifnb1- Ifnb1* L2 enhancer) 102.3 nm between 561 and 488 nm (*Il12b - Il12b* HSS1 enhancer and *Peli1-Peli1* enhancer), and 41.4 nm between 561 and 633 nm (*Prdm1-Prdm1* enhancer). In each case the chromatic aberration was smaller than the xy pixel size of 189 nm, and the data has not been corrected for chromatic aberration. We have added this information to the methods section of the revised manuscript.

Finally the BET inhibitor findings are very interesting but again it is possibly over simplified in the abstract and discussion. Figure 8b is slightly confusing and might be helped if the Y axes were on the same scale. It appears that the BET inhibitors and Rad21 knockout have a different effect at *Egr2* (where the eRNAs are reduced) compared to *Il12b*, where they don't

change. It would be interesting if the authors could comment on this. It appears to be a complex phenomenon and the statements ‘Cohesin is largely dispensable for the regulation of enhancer burst frequencies.’ and ‘Our data ... suggest that cohesin and BD2 act on a shared pathway.’ are probably overstated.

The wording "Cohesin is largely dispensable for the regulation of enhancer burst frequencies" reflects our observation that most of the enhancers we have tested show similar probabilities of bursting in LPS-activated control and Rad21 ko macrophages.

We agree with the referee that the BET inhibitor findings - although novel and interesting - are currently less nuanced compared to the remainder of the manuscript. We have moderated our statement in the abstract to "Our data ... suggest the possibility that cohesin and BD2 may act on a shared pathway".

In Figure 8b we have chosen to present the wild-type and Rad21 ko data on different scales because the fraction of bursting alleles is much reduced in the absence of cohesin. Referee 3 Figure 2 shows a version of the same figure where burst fractions are plotted relative to the LPS control (in the absence of BD inhibitors). We hope this conveys the point that BD2 reduces the burst fractions of *Egr2* and *Ii12b* in wild-type macrophages, but has no effect on the burst fractions of *Egr2* and *Ii12b* in cohesin-deficient macrophages.

Referee 3 Figure 2. Perturbations of cohesin and the second BET bromodomain BD2 have non-additive effects on burst fractions.

Transcriptional burst fractions of *Egr2* and *Ii12b* in wild-type and cohesin-deficient macrophages 1h after LPS activation in the presence of the BD1-selective inhibitor GSK778 (1mM), the BD2-selective inhibitor GSK046 (1mM), or carrier (DMSO) represented as a fraction of the median value in LPS in the absence of BD inhibitors. Boxes show upper and lower quartiles and whiskers show 1.5 of the interquartile range. Statistical test for burst fraction: Cochran-Mantel-Haenszel test (replicate as strata) with Bonferroni correction for multiple testing, 2 sided. For *Egr2*, 12032 wild-type and 9097 *Rad21*^{-/-} cells were analysed in 3 independent biological smRNA-FISH replicates. For *Ii12b*, 28863 wild-type and 23155 *Rad21*^{-/-} cells were analysed in 3-5 independent biological smRNA-FISH replicates.

We hope that these changes go some way towards addressing the referee's concerns.

Reviewer #4 (Remarks to the Author):

The paper by Robles-Rebollo focuses on the control of the inflammatory responses via transcriptional bursting using an impressive range and state-of-the-art single cell and population-level genomics and imaging approaches. The authors demonstrate that the control of the inducible gene expression in the TLR system is achieved by frequency modulation of transcription. Although this has been suggested before, the novel and important contribution of this paper is to link this transcriptional regulation to cohesin-mediated enhancer bursting. The most elegant part of the paper demonstrates how the enhancer bursting couples with promoter bursting (using a dual enhancer and intronic smRNA-FISH) and uses a mice model to demonstrate that cohesion regulates this coupling. The actual physical model of how cohesion achieves transcriptional control is not provided, but the authors use small molecules inhibitors to demonstrate that cohesion acts with BET proteins, but is unrelated to DNA damage.

Overall the manuscript includes an overwhelming amount of data, analyses are performed to a very high standard, while the figures and text are well written. I believe that the paper provides a step-change in our understanding of the regulation of gene expression in general and in the case of LPS-induced responses.

We thank the referee for the positive overall assessment of our study.

The authors use nascent smFISH-mRNA as the key readout for the burst size, and a fraction of actively transcribing alleles as bursting frequency. This methodology requires further validation. For example, the burst size, the quantity of mRNA produced in the single burst (kt/koff in the random telegraph model), depends on the duration of the burst, not just on the transcription rate (as provided by the number of mRNA molecules at the transcription site, i.e.). In turn, the fraction of actively transcribing alleles does not provide insight into the dynamical regulation, i.e., are we simply looking at one switch per allele (OFF->ON) upon activation, or indeed we have a transcriptional bursting process involving multiple switches of the same allele. Given that some data describes the very initial part of the response (up to 2h), which may indeed correspond to just a single switch, while other data extend to 8h of stimulation, this might change the context of the “bursting” described in the manuscript. The temporal aspect is also relevant for coupling between enhancer and promoter bursting, where the apparent differences in fractions could be due to underlying kinetics and different switching.

We agree with the referee that the points raised here are important, and have revised the manuscript accordingly.

First, the revised manuscript now clearly states that we use smRNA-FISH for the quantification of transcript copy numbers and transcriptional bursts in thousands of individual macrophages during the first 2h of the response to LPS. **"During this transient response, the expression of many inducible immune genes is biphasic (Shalek et al. 2014), indicating that**

only a subset of cells activate the expression of a given gene during any one activation cycle. In this setting, cell to cell variability likely reflects the probability of a gene turning on at least once during the activation cycle, and the smRNA-FISH burst fraction captures the probability that alleles initiate transcription"

Second, the revised manuscript acknowledges that our smRNA-FISH approach measures parameters that are distinct from burst sizes and burst frequencies as inferred mathematically from the moment of mRNA distributions. The revised manuscript describes the data in terms of the fraction of bursting alleles and the intensity of transcriptional bursts. The relevant section of the revised results section now reads: " **Nascent smRNA-FISH captures the fraction of actively transcribing alleles and the intensity of probe signal at transcription start sites (Burst intensity, Fig.1c). We found that the inducible expression of *Ii12b* (Fig.1d) and other inducible genes tested by smRNA-FISH (*Egr2*, *Prdm1*, *Ifnb1*, *Peli1* and *Sertad2*, Supplementary Fig. 2) was associated primarily with an increase in the fraction of actively transcribing alleles"**.

Third, to allow comparisons with previously reported parameters we have also added calculations of moment-derived burst frequency and burst size. The revised manuscript describes the results of this analysis as follows: "This indicates that burst fraction and burst frequency are correlated in LPS-activated macrophages' (Fig. 1e, left). In contrast, burst intensity as quantified by smRNA-FISH poorly correlated with burst size as inferred by the moment of mature mRNA copy number distribution (Fig.1e, right)".

We consider that the poor correlation between burst intensities and burst sizes may be related to the nature of inducible immune gene expression during the LPS response of macrophages, which is transient, not steady-state. Inducible immune gene expression involves a chain of hierarchical events that changes mRNA distributions dynamically, and therefore, these distributions cannot be assumed to be in steady-state, particularly at early time points (Bhatt et al., 2013).

Revised Fig. 1e. Comparison of burst fractions with burst frequencies, and of burst intensities with burst sizes.

e) Left: correlation between the fraction of actively transcribing alleles measured by smRNA-FISH for *Il12b*, *Ifit1*, *Peli1*, *Cxcl10*, and *Ifnb1* and transcriptional burst frequencies ($f_m = \mu / (b_m - 1)$) inferred from the moments of mature transcript distributions in macrophages stimulated with LPS for 90 min (*Il12b*, *Ifnb1*, *Peli1*), 120 min (*Cxcl10*) and 180 min (*Ifit1*). Right: correlation between burst intensity measured by smRNA-FISH and burst size ($b_m = \sigma^2 / \mu$) inferred mathematically from the mean μ and the variance σ^2 of the distribution of mature transcripts. Intron probes were used for *Il12b* and *Peli1*, exon probes were used for

Ifit1, *Cxcl10*, and for the intron-less *Ifnb1* gene. r = Pearson correlation. P -value derive from Pearson's product-moment correlation test as implemented in the `cor.test` function in R against the null hypothesis correlation is equal to 0. A total of 18494 cells were analysed in 3-4 independent biological replicates per transcript.

Other comments:

Fig. 1e legend is confusing, please clarify how normalisation was performed. Are matching RNA-seq time points available? Not clear from the description.

Fig. 1e has been revised as detailed above

Fig. 1f. Most of the analyzed genes appear to have very low abundance, except of the first and last clusters (not clear if they correspond to A-to-F from the cited reference). Now authors use a different measure to demonstrate the dominance of burst frequency. The comparison with previously used measures should be shown, e.g. do burst size and frequency calculated using moments for this scRNA-seq data show the same trend, does smRNA-FISH data formatted as in f show this trend, do individual genes from b (e.g. *Il12b*) show expected trends?

We agree with the referee and have revised our scRNA-seq analysis to include only those inducible genes that are expressed by at least 20% of cells in at least one condition. The resulting Figure 1f is shown below. The conclusions remain unchanged.

Revised Figure 1f. Left: heatmap of transcript levels (CPM) of inducible macrophage primary and secondary response genes (Bhatt et al., 2012) detected by scRNA-seq in at least 20% of cells. Centre: Fraction of cells with detectable transcripts for each of 6 previously defined classes of inducible genes (Bhatt et al. 2012) at 0, 2 and 8h after LPS activation. Right: Transcript levels (CPM+1) per transcript-positive cell is shown for the same classes of inducible genes at 0, 2 and 8h after LPS activation. Boxes show upper and lower quartiles and whiskers show 1.5 of the interquartile range. Numbers represent adjusted P -values for 0h versus 2h and 0h versus 8h. Wilcoxon signed rank test with Bonferroni correction for multiple testing. One experiment with 682 cells.

Fig. 2 b, burst size calculation relies on the size of the mature mRNA molecule, not clear from the images if they are produced. Please clarify.

As detailed above, the revised manuscript acknowledges that our smRNA-FISH approach measures parameters that are distinct from burst sizes and burst frequencies as inferred mathematically from the moment of mRNA distributions. The revised manuscript describes the data in terms of the fraction of bursting alleles and the intensity of transcriptional bursts. The relevant section of the revised results section now reads: "**Nascent smRNA-FISH captures the fraction of actively transcribing alleles and the intensity of probe signal at transcription start sites (Burst intensity, Fig.1c). We found that the inducible expression of *Il12b* (Fig.1d) and other inducible genes tested by smRNA-FISH (*Egr2*, *Prdm1*, *Ifnb1*, *Peli1* and *Sertad2*, Supplementary Fig. 2) was regulated primarily by an increase in the fraction of actively transcribing alleles**".

Fig. 3. Enhancer bursting is more efficient than transcriptional bursting (~2-fold higher difference in fraction of active cells, but lower burst size), suggesting that many enhancers bursts are not productive or not coordinated. These differences might be related to the underlying dynamics of the process, i.e. the frequency of multiple ON-OFF switching... This information can be extracted by analyzing smRNA-FISH data with telegraph models and should provide additional insight.

We agree that the higher burst fraction of enhancers compared to promoters is interesting. We may not have understood the referee correctly, but it would seem that the inherent instability of eRNAs would complicate an analysis based on eRNA transcript distributions. We are limited to evaluating eRNA transcription by 2 approaches (i) GRO-seq at the population-level and (ii) the direct observation of transcription sites by smRNA-FISH where we measure the fraction of bursting enhancer alleles (i.e. the burst fraction), and the signal intensity at these enhancers (i.e. the burst intensity).

The presented data highlight the correlation between the enhancer and promoter bursting. However, does not indicate whether intron bursts occur without the corresponding enhancer bursting. This would be important to visualize for completeness. As above, does the absence of enhancer sequence completely abolish transcription from the locus, or just reduce the fraction/burst size of responding cells? While validation of *Il12b* enhancers has been achieved previously using population-level analyses, the single-cell interpretation remains unclear. Functional validation of enhancers would be an important addition to the manuscript, although this can be technically difficult.

The referee is right that - despite the observed coordination of enhancer and promoter bursting - promoter bursts also occur in the absence of enhancer bursts, and vice versa (Referee 4 Figure 1.). This is in agreement with a growing literature indicating that the occurrence of enhancer and promoter bursts is not strictly linked in time or space (Huang et al., 2021 Nat Genet. 53: 1064-74; Alexander et al., 2019, eLife 8, e41769; Benabdallah et al. 2019 Mol. Cell 76, 473–84; Fukaya et al., 2016 Cell 166, 358-68).

Referee 4 Figure 1. The fraction of cells with *I12b* promoter bursts and/or *I12b* HSS1 enhancer bursts is shown for 4 biological replicates 90 min after LPS stimulation of wild-type macrophages.

Fig. 4. scRNA-seq data appears to be very sparse for the majority of the genes. It appears that only clusters presented at the top and bottom are robustly expressed (as in Fig 1f). Are we seeing a reduced number of drop-out events (due to sequencing coverage) as the expression increases in stimulated cells? Removing genes that are not robustly expressed should make this analysis more stringent.

We agree and have revised our scRNA-seq analysis to include only those inducible genes that are expressed by at least 20% of cells in at least one condition. The resulting Figure 4c is shown below. The conclusions remain unchanged.

Revised Figure 4c. Left: heatmap of inducible macrophage genes (Bhatt et al., 2012) detected by scRNA-seq in at least 20% of wild-type (grey) or *Rad21*^{-/-} macrophages (red). Middle: Fraction of cells with detectable transcripts for inducible genes. Right: The expression level of LPS-inducible transcripts in cells with detectable transcripts is plotted as $\ln(\text{CPM} + 1)$. Boxes show upper and lower quartiles and whiskers show 1.5 of the interquartile range. Numbers represent adjusted *P*-values. Wilcoxon signed-rank test with Bonferroni correction for multiple testing. One experiment with 1362 cells.

A recent paper analyzed the role of burst frequency and size in the TLR-mediated responses in macrophages, see (Bagnall et al. 2020), while others use similar modelling methodology to analyse control of NF- κ B transcription, e.g. (Bass et al. 2021). These should be acknowledged in the manuscript.

We thank the referee for pointing out these papers, which we reference in the revised manuscript: "Transcription is typically discontinuous, and can be described in terms of the

frequency and the size of transcriptional bursts (Golding et al 2005, Chubb et al 2006, Raj et al 2006, Suter et al 2011). Both the size (Falo-Sanjuan et al. 2019; Lee et al. 2019; Stavreva et al., 2019; Bagnall et al., 2020; Bass et al., 2021) and the frequency (Senecal et al., 2014; Hendy et al., 2017; Dar et al., 2017; Chen et al., 2019; Stavreva et al., 2019; Bagnall et al., 2020) of transcriptional bursts have been linked to inducible gene expression in different experimental systems".

And: "We compared the fraction of actively transcribing alleles, as directly quantified by smRNA-FISH, with and the burst frequency as inferred mathematically from the moments (the mean and the variance) of mature mRNA distribution (Fig.1e, left; So et al., 2011; Raj et al., 2006; Bagnall et al 2200; Bass et al., 2021)".

We hope that these changes go some way towards addressing the referee's concerns.

REVIEWERS' COMMENTS

Reviewer #2 (Remarks to the Author):

The authors have addressed my questions and I continue to think that this is a very nice paper that makes an important contribution to the field.

Reviewer #3 (Remarks to the Author):

We would like to thank the authors for carefully addressing all of the points that we raised during the first round of review. We have no further comments and would be very supportive of publication at this stage.

Reviewer #4 (Remarks to the Author):

The authors addressed my previous comments. Recommend the manuscript for publication